# TPP-Based Nanovesicles Kill MDR Neuroblastoma Cells and Induce Moderate ROS Increase, While Exerting Low Toxicity Towards Primary Cell Cultures: An In Vitro Study

**DOI:** 10.3390/ijms26114991

**Published:** 2025-05-22

**Authors:** Silvana Alfei, Carola Torazza, Francesca Bacchetti, Marco Milanese, Mario Passalacqua, Elaheh Khaledizadeh, Stefania Vernazza, Cinzia Domenicotti, Barbara Marengo

**Affiliations:** 1Department of Pharmacy, University of Genoa, 16148 Genoa, Italy; carola.torazza@unige.it (C.T.); francesca.bacchetti@edu.unige.it (F.B.); marco.milanese@unige.it (M.M.); 2Scientific Institute for Cancer Research (IRCCS), Ospedale Policlinico San Martino, 16132 Genova, Italy; barbara.marengo@unige.it; 3Department of Experimental Medicine (DIMES), University of Genova, 16132 Genoa, Italy; mario.passalacqua@unige.it (M.P.); elaheh.khaledizadeh@edu.unige.it (E.K.); stefania.vernazza@unige.it (S.V.); 4Centro 3R, Department of Information Engineering, University of Pisa, 56122 Pisa, Italy

**Keywords:** high-risk neuroblastoma (HR-NB), HTLA ER cells, triphenyl phosphonium-bola amphiphile nanovesicles, spinal cord astrocytes, cortical neurons

## Abstract

Neuroblastoma (NB) is a malignant childhood tumour, which originates from neuroblasts with an incidence of approximately 15,000 new cases per year worldwide. Therapy-induced secondary tumorigenesis and the emergency of drug resistance in its high-risk (HR-NB) forms drive to a survival rate of <50%, despite aggressive treatments. Our recent research is focused on testing in vitro the effects of synthetized triphenyl phosphonium (TPP)-based bola amphiphilic nanovesicles (BPPBs) against both drug-sensitive and multi-drug-resistant (MDR) cancer cell lines. In the present study, BPPB demonstrated sub-micromolar IC_50_ values (0.4–0.9 µM) towards drug-sensitive HTLA 230, while 1.20–1.35 µM IC_50_ were determined on MDR HTLA ER. Noteworthily, we have demonstrated that BPPB triggers apoptosis of both NB cell populations. Additionally, since MDR NB cells (HTLA ER) are equipped with higher levels of antioxidants than sensitive ones (HTLA 230), the potential involvement of reactive oxygen species (ROS) in the cytotoxic action of BPPB was also investigated. Then, a novel analytical approach was applied to the results of cell viability and ROS monitoring for their better interpretation. Proper dispersion graphs and their best fitting nonlinear regression models were used to verify if the cytotoxic effects of BPPB could depend on BPPB concentrations, exposure times, and/or ROS generation, and if ROS increase could depend on BPPB concentrations and/or exposure times. A ROS-dependent mechanism was found in 24 h and 24/48 h treatments of HTLA ER and HTLA 230, respectively. Furthermore, the potential clinical development of BPPB as a new curative option for children affected by HR-NB was assessed by testing BPPB on astrocyte and neuron primary cell cultures, and analytical correlation studies were used to interpret the results. Notably, BPPB administration was sufficiently and well tolerated by neurons and astrocytes, respectively, allowing selectivity index values of up to 23.7. These in vitro results, associated with the low haemolytic activity of BPPB, pave the way for future in vivo investigations and, upon confirmation, for the possible development of BPPB as a novel therapeutic strategy to treat MDR HR-NB.

## 1. Introduction

Neuroblastoma (NB) is a malignant childhood solid tumour originating from the neuroblast cells present in the sympathetic nervous system [1,2]. Neuroblasts are immature or developing cells found in nerves, which are spread in the body, including neck, adrenal glands, chest, and spine [3]. For this reason, NB can arise in different body parts, including the adrenal glands, which are located above the kidneys, or in the nerve ganglia in the chest and abdomen [3]. NB represents 10% of all paediatric tumours, and frequently, at the time of diagnosis, it is already characterized by metastatic spread to the skeleton and bone marrow, making it particularly refractory also to extensive multiple-components chemotherapy [4]. The overall incidence of NB in children up to 15 years of age is approximately 15,000 new cases per year worldwide, causing 15% of childhood cancer-related mortality [1,2]. Therefore, new anticancer approaches are needed to improve the life expectancy of NB patients. Current advances in therapy are successful in low- and intermediate-risk NB patients, with a survival rate of up to 70%. On the contrary, less than 50% of high risk (HR) patients achieve long-term survival, highlighting the urgent need to find new more effective treatments. Moreover, 44% of survivors experienced late morbidity, while tumour recurred in 11.5% of patients, who must be monitored for tumour recurrence and long-term sequelae. Since genetic and epigenetic alterations are among the main factors contributing to the pathogenesis and refractoriness of HR-NB, targeted therapeutic approaches have been developed. In this context, the current targeted therapies include those pointing genetic aberrations, disrupted signalling molecules, as well as norepinephrine and somatostatin receptors by radiopharmaceuticals [5]. Furthermore, it has been observed that patients undergoing immunotherapy have a better prognosis and a greater therapeutic response to drugs administered at doses lower than those prescribed by the therapeutical protocol. For this reason, immunotherapy has been now included in first-line treatment protocols. Unfortunately, despite the addition of monoclonal antibodies, the survival rate for stage IV HR-NB remained at 40–50% [6], thus indicating that other molecular targets are urgently needed to efficiently treat this disease [7].

Moreover, this dramatic condition is further complicated by the onset of therapy resistance, which induces tumour recurrence and is responsible for increasing cancer-related mortality [8]. Drug resistance is due to the adaptive genetic mechanisms acquired by cancer cells, which lead to drug inactivation, the alteration of drug targets, and changes in metabolic and apoptotic processes [9]. In this regard, a combined therapeutic approach could delay the onset of resistance and improve patient outcomes. However, undesired drug–drug interactions could negatively affect the therapy effects [10]. Therefore, the development of compounds with an extra-genomic mechanism of action could be useful to overcome the mutational mechanisms, by which tumour cells acquire drug resistance [11], offering new therapeutic opportunities capable of eradicating or, at least, reducing the tumor and increasing the quality and the life expectancy of patients with NB. In this context, we have recently demonstrated that triphenyl phosphonium (TPP)-based bola amphiphilic nanovesicles (BPPBs), synthetized by us, are strongly cytotoxic to both drug-sensitive and -resistant NB and cutaneous metastatic melanoma (CMM) cells [11,12,13]. The details of these early investigations are available in Appendix A. In this new study, we have investigated more in depth the possible mechanisms underlying the cytotoxic effects of BPPB observed on both drug-sensitive (HTLA 230 human stage-IV) and multidrug-resistant (MDR) (HTLA ER) NB cells. To this end, reactive oxygen species (ROS) production was monitored, while experiments to assess apoptosis and necrosis in both cell populations were carried out. Noteworthily, an innovative analytical approach was used to better interpret the results of cancer and normal cell viability. Specifically, we have analytically assessed if the cytotoxic effects of BPPB could depend on BPPB concentrations, the exposure time, and/or the ROS increase. The study was then extended also to the results of ROS monitoring, thus assessing if the ROS increase could depend on exposure timing and/or BPPB concentrations. To this end, proper dispersion graphs and their best fitting regression models were used to analyse several series of data. With this approach, we have really established the existence or absence of a certain correlation between different couples of data, thus confirming or rebutting the presence of dependence between them. Furthermore, to better support the potential clinical development of BPPB as a new strategy to cure children with HR-NB, the effects of BPPB nanovesicles on spinal cord astrocyte and cortical neuron primary cell cultures were for the first time tested.

## 2. Results and Discussion

### 2.1. 1,1-(1,12-Dodecanediyl)-bis-[1,1,1]-triphenyl-phosphonium Di-Bromide (BPPB)

The alkyl tri-phenyl-phosphonium derivative displaying two triphenyl cationic moieties linked to each other by a C12 alkyl chain (BPPB), used here for more in-depth investigations on drug-sensitive NB cells and on their multi-drug-resistant counterpart, was synthetized performing the procedure recently described [14].

### 2.2. Cytotoxic Effects of BPPB Towards HTLA 230 and HTLA ER NB Cells

In this section, the strong anticancer effect of BPPB against ETO-sensitive HTLA 230 and multidrug-resistant (MDR) HTLA ER neuroblastoma (NB) cells previously reported was confirmed [11]. Cell viability in HTLA 230 and HTLA ER cells exposed to increasing concentrations of BPPB for 24, 48, and 72 h was assessed by using the MTS essay, which is an efficient, eco- and user-friendly tool, commonly used to investigate the behaviour of cancer cells when exposed to specific treatments and how these treatments influence their growth and viability [15].

As shown in Figure 1a (HTLA 230 cells), BPPB caused a significant reduction in cells’ viability, starting from a 0.5 µM concentration after 24 h treatment, while from 0.1 µM and 0.25 µM after 48 and 72 h exposure, respectively. Cell viability decreased under 50% for BPPB concentrations ≥ 0.75 µM after 24 h (42.5%), ≥0.5 µM after 48 h (41.8%), and ≥0.25 µM after 72 h (47.6%). Once under 50%, cells’ viability was similar for treatments of 24 (39.3–45.3%) and 48 h (36.2–41.9%), and for all other concentrations tested, thus demonstrating a cytotoxic mechanism probably not dependent on increasing BPPB concentrations. On the contrary, when treated with BPPB for 72 h, the viability of cells decreased from 47.6% to 21.7%, with a probable concentration-dependent trend. Concerning HTLA ER cells, they were more tolerant to BPPB than HTLA 230, as previously observed [11]. Particularly, BPPB caused a significant reduction in the viability of cells at concentrations ≥ 0.75 µM after 24 h and 48 h treatments, while at concentrations ≥ 1.00 µM after 72 h exposure (Figure 1b). Cell viability decreased under 50% for BPPB concentrations ≥ 1.25 µM after 24 h (42.5%) and 72 h (39.8%) and ≥1.5 µM after 48 h (39.4%). The cytotoxic effects of BPPB vs. HTLA ER cells were similar for treatments of 24 and 72 h, while they were minor for treatments of 48 h, thus foreseeing the absence of a linear correlation between the cytotoxic effects of BPPB and exposure timing. Specifically, at the highest BPPB concentration tested (2 µM), cell viability was 26.5% (24 h), 33.1% (48 h), and 25.8% (72 h) (Figure 1b).

To calculate the IC_50_ of BPPB against both cell lines, the bar graphs of Figure 1a,b were converted into the dispersion graphs reported in Figure 1c,d.

Following this, µM concentrations (x) were transformed in Log_10_ (x) (Figure 1e (HTLA 230) and Figure 1f (HTLA ER), traces with indicators and error bar). The nonlinear regression models of the Log_10_ (BPPB concentrations) vs. the normalized response (Hill Slope) shown in Figure 1e (HTLA 230) and Figure 1f (HTLA ER), traces without indicators, elaborated by GraphPad Prism software version 8.0.1 (GraphPad Software, Boston, MA, USA), gave us the IC_50_ values of BPPB for both cell populations at 24, 48, and 72 h of treatment. The numerical results ± S.D. are reported in Table 1.

The results in Table 1 confirm the trend observed in Figure 1, establishing a higher tolerance of HTLA ER cells with respect to HTLA 230 cells. Specifically, the cytotoxic effects of BPPB vs. HTLA 230 were higher by 25.3% (24 h), 65.9% (48 h), and 67.5% (72 h) than those observed vs. HTLA ER cells. Despite this higher tolerance to BPPB treatments, in experiments carried out for 24 h on multidrug-resistant HTLA ER, the IC_50_ value of BPPB was 478-fold lower than that determined with ETO in the same conditions, due to the acquired resistance to several available chemotherapeutics that emerged in such cells [16]. According to these results, BPPB was 64–86-fold more cytotoxic to both the cell lines considered here than the titanium oxide-Tantalum (TiO_2_-Ta) nanoparticles (NPs) reported by Almutairi et al. While the authors observed an IC_50_ value = 68 µg/mL (79.7 µM) to human neuroblastoma (SH-SY5Y) cells, after 24 h exposure, the IC_50_ values of BPPB after the same time were 0.9257 µM (HTLA 230) and 1.2390 µM (HTLA ER), respectively [17]. Similarly, BPPB was more cytotoxic by 4–30 times towards the HTLA 230 and HTLA ER cell lines than the silver nanoparticles (AgNPs) reported by Sambale et al. [18]. In the study, the authors observed IC_50_ values = 10, 6, and 4 µg/mL (11.7, 7 and 4.7 µM) towards A-549, HEP G2, and PC-12 cells, respectively, after 72 h exposure, while the IC_50_ values of BPPB after the same time were 0.3892 µM (HTLA 230) and 1.1990 µM (HTLA ER), respectively [18]. Haase et al. reported an IC_50_ value of 110 µg/mL (129 µM) for peptide-coated AgNPs of 20 nm (Ag20Pep) and of 140 µg/mL (164 µM) for Ag40Pep, against THP-1 leukemia cell lines, after 24 h exposure. After 48 h exposure, the decrease in cells’ viability became much steeper, with IC_50_ values of 18 µg/mL (21 µM) (Ag20Pep) and 30 µg/mL (35 µM) (Ag40Pep) [19]. According to these results, after 24 h exposure, BPPB was more potent by 105–143 times if Ag20Pep is considered, and by 132–182 times if Ag40Pep is considered. Despite the difference in potency being lower after 48 h exposure, BPPB was anyway more potent than Ag20Pep by 18–54 times and Ag40Pep by 29–90 times [20]. Moreover, Skrzypczak et al. modified geldanamycin (GDM) using different types of neutral/acidic/basic substituents, quinuclidine motif, and then transformed the obtained products into ammonium salts at C (17) [21]. Ammonium salts, namely compounds **9**–**13** in the work, demonstrated an attractive cytotoxic potency against the MCF-7 breast cancer cell line (IC_50_∼2 µM) after 72 h exposure. Also in this case, BPPB displayed a cytotoxicity 1.7–5.1-fold higher than that of compounds **9**–**13** developed by Skrzypczak et al. The use of compound **13** in association with potentiators, including PEI and doxorubicin (DOX), enhanced its anticancer effects towards SKBR-3, SKOV-3, and PC-3 cancer cells, lowering the IC_50_ values from 2 µM to ∼0.5 µM [21]. This new value was anyway higher than that observed by us with BPPB against HTLA 230 (0.3892 µM), without recovering to the use of potentiators. Also, Nene and co-authors monitored the therapeutic activity of new triphenyl-phosphonium-labeled phthalocyanines (Pcs), namely 2,9,16,23-tetrakis(*N-*(*N*-butyl-4-triphenyl-phosphonium)-pyridine-4-yloxy) Zn(II) Pc (**3**) and 2,9,16,23-tetrakis-(*N-*(*N*-butyl-4-triphenyl-phosphonium)-morpholino) Zn(II) Pc (**4**) upon exposure to light (photodynamic therapy, PDT), ultrasound (sonodynamic therapy, SDT), and the combination of light and ultrasound (PDT/SDT) against MFC-7 and Hela cancer cells [22]. The IC_50_ values of **3** against MFC-7 and Hela cells, after 24 h exposure, were 43.3 (PDT), 18.2 (SDT), and 13.2 (PDT/SDT) µM, and 21.8 (PDT), 14.3 (SDT), and 12.7 (PDT/SDT) µM, respectively [22]. Conversely, the IC_50_ values of **4** against MFC-7 and Hela cells, after 24 h exposure, were 45.0 (PDT), 20.5 (SDT), and 15.5 (PDT/SDT) µM, and 25.6 (PDT), 17.2 (SDT), and 14.4 (PDT/SDT) µM, respectively [22]. Regarding these results, considering the best result obtained by the authors regarding compound **3** associated with the combination PDT/SDT, the anticancer effects of BPPB observed by us were higher by 10.3–14.7 times. In all other cases, the superiority of BPPB versus compounds **3** and **4** was even more evident, without the need to help it with light, sonication, or both. Collectively, these in vitro results establish the superiority of BPPB in fighting cancer, compared to several other nanoparticles previously tested against different tumour models, and pave the way for future in vivo investigations.

#### 2.2.1. BPPB Induces Early and Late Apoptosis in Neuroblastoma Cells

Although it has been reported that diverse therapeutic approaches to counteract cancer, including chemotherapeutic drugs, gamma irradiation, suicide genes, or immunotherapy, are mediated through the induction of apoptosis [23], also necrosis, as well as other forms of cell death, that are not classified so far, may be triggered by anticancer therapies [23]. Here, to evaluate if BPPB induced apoptosis or necrosis, both NB cell populations were labelled with Annexin-V-FITC/propidium iodide (PI) and analysed by confocal fluorescence microscopy, as reported in the literature [24,25,26]. This is a standard procedure to monitor the progression of apoptosis and the possible presence of necrosis [27,28,29,30,31]. In this regard, to investigate the modality of BPPB-induced cell death, HTLA 230 and HTLA ER cells were treated for 24 h with 0.5 µM and 1.0 µM BPPB, respectively, and then labelled with 4′,6-diamidino-2-phenylindole (DAPI), a fluorescent dye able to stain only the nuclei of dead or dying cells, where the membrane is compromised [32]. As shown in Figure 2, according to these early experiments, BPPB seemed to induce the necrosis of HTLA 230 cells, as demonstrated by the highest DAPI labelling shown by treated cells in respect to the control ones. On the contrary, BPPB treatment did not induce the necrosis of HTLA-ER cells.

However, to further clarify the mechanism of cell death induced by BPPB, NB cells were labelled with Annexin V-FITC and propidium iodide, as markers of apoptosis and necrosis/late apoptosis, respectively. As shown in Figure 3, BPPB caused a marked apoptosis of both treated cell populations. However, in HTLA 230 cells, this effect was induced by a dose of BPPB 50% lower than that used to treat HTLA ER cells. In addition, a high number of HTLA 230 labelled by Annexin was also labelled by PI, demonstrating that the necrosis, identified by DAPI staining (Figure 2), was a late effect of the apoptotic death.

#### 2.2.2. Correlation Study

The simple analysis of Figure 1a–f does not provide any evidence on the factors by which BPPB exerts its cytotoxic effects, including the concentration, exposure time, or reactive oxygen species (ROS) overproduction. The same goes for the increase in ROS reported later, and ultimately for the cytotoxic effects of BPPB on the primary cells reported later. A concentration-dependent, time-dependent, or ROS-dependent effect/mechanism has been often reported, without however verifying the accuracy of these assumptions analytically. Therefore, by a novel analytical approach, we have carried out an innovative study to assess if the cytotoxic effects of BPPB to cancer and normal cells could depend on BPPB concentrations, exposure time, and/or ROS increase. Such study was then extended also to the results of ROS monitoring to assess if the ROS increase could depend on exposure timing and/or BPPB concentrations. To this end, proper dispersion graphs have been constructed, and their best fitting regression models were used to analyse several series of data. With this approach, we have really established the existence or absence of a certain correlation between different couples of data, thus confirming or rebutting the presence of dependence between them.

##### Correlation Between BPPB Cytotoxic Effects and BPPB Concentrations

The dispersion graphs of the cell viability (%) of both HTLA 230 and HTLA ER after 24, 48, and 72 h of exposure to increasing BPPB vs. BPPB concentrations of 0.1–2.0 µM, and their best fitting statistical models were used to establish the existence of a concentration-dependent mechanism for the cytotoxic effects of BPPB. Following a method recently reported [13], we explored the existence or absence of a significant correlation between BPPB anticancer effects and its administered concentrations, based on the R^2^ values of the statistical models which best fitted the data of dispersion graphs. The literature’s indications about the significance of R^2^ and its possible limitations when applied to nonlinear models were considered, before accepting the results [33,34,35]. The nonlinear regression models and all associated parameters were achieved using Microsoft Excel software 365. For cross-validation, according to the literature, the overall accuracy of the models was investigated by analyzing the prediction power of the best nonlinear models, selected early based on the R^2^ values [35]. To this end, we plotted the values predicted by models using their equations against observed ones. A well-fitting model should have points clustered closely around a 45-degree line, whose linear regression should show high values of R^2^ (>0.95) [35]. Appendix A shows the dispersion graphs of the cell viability of HTLA 230 cells (red round (24 h), pink square (48 h), and light pink triangular (72 h) indicators) and of HTLA ER (purple round (24 h), blue square (48 h), and sky blue (72 h) indicators) cells vs. BPPB concentrations (0.1–2.0 µM). In Appendix A, the nonlinear regression models (dotted lines in the same colors of indicators), which best fit the data of dispersion graphs according to R^2^, have been reported. The equations and values of R^2^ related to the selected models have been shown in the same colors of models. Appendix A shows the plots of the values predicted by models using their equations against the observed one. The related linear regressions, their equations, and R^2^ values have been also shown using the same colors of indicators. According to the results in Appendix A, the R^2^ values obtained for the linear regressions reported in Appendix A were identical to those obtained for the nonlinear models shown in Appendix A, thus establishing that cross-validation is a redundant experiment, which can be avoided, and it was not performed further. Values of R^2^ > 0.95 (0.9575 and 0.9741) were obtained only for the statistical models used to describe the data of HTLA ER cell viability vs. BPPB concentrations after 24 and 48 h treatments, establishing the existence of a good polynomial correlation of second order between data. Values of R^2^ close to 0.95 (0.9466, 0.9327) were obtained for the statistical models selected to describe the same data of HTLA 230 after an exposure timing of 72 and 24 h, establishing the existence of an inferior correlation of Power type and of a second-order polynomial type between data, respectively. R^2^ values of 0.9250 (72 h) and 0.8981 (48 h) were finally obtained for the statistical models chosen to describe the data of HTLA ER and HTLA 230 cells, respectively, thus establishing a very poor polynomial correlation of second order between them. This investigation evidenced that the anticancer effects of BPPB are dependent on its concentrations only for HTLA ER cells when treated for 24 and 48 h. When these cells were treated for a longer period of 72 h, the dependence of BPPB cytotoxic effects on its concentrations was scarce. On the contrary, concerning HTLA 230 cells, the dependence of BPPB cytotoxic effects on its concentrations was appreciable at the highest exposure time of 72 h, absent at 48, and scarce at 24 h of treatments.

##### Correlation Between BPPB Cytotoxic Effects and Exposure Time

The possible existence or absence of any type of dependence of BPPB anticancer effects expressed as IC_50_ values on the times of treatments were investigated by a method like that used in the previous section. The nonlinear regression models and all associated parameters were achieved using Microsoft Excel software 365. After having tried different linear and nonlinear mathematical models, based on the very high R^2^ values (R^2^ = 1), it was found that the model which best describes the dispersion graphs of the IC_50_ values (µM) vs. the times of exposure (hours) got for the HTLA 230 cells (red round indicators) and HTLA ER ones (purple square indicators) is of a second-order polynomial type, thus assessing the existence of a strong second-order polynomial correlation between BPPB cytotoxic effects and time of exposure (Appendix A). The found correlation established the existence of a time-dependent cytotoxic mechanism for the effects of BPPB on both NB cell populations. These findings were in line with those recently observed when BPPB was tested on MeOV and MeTRAV PLX-R cells [13]. Additionally, multidrug-resistant HTLA ER cells were 1.3, 2.9, and 3.1 times more tolerant to BPPB than HTLA 230 ones, at 24, 48, and 72 h of exposure, respectively (Appendix A).

### 2.3. BPPB Induces ROS Overproduction in HTLA 230 and HTLA ER Cells: Concentration- and Time-Dependent Experiments

It has been reported that the administration of TPP-based compounds, including TPP-based bola amphiphilic nanovesicles such as BPPB, induce ROS overproduction, thus causing oxidative stress (OS) [12,36], which has been widely reported to be involved in the anticancer effect of several chemotherapeutic drugs, including etoposide [37], doxorubicin [38], and cisplatin [39]. In this context, our recent studies demonstrated the role of a ROS-dependent mechanism underlying the anticancer effects of BPPB on two PLX-sensitive BRAF-mutant CMM cells [12]. On the contrary, the role of ROS was scarcely present in PLX-resistant CMM cells exposed to the same concentrations of BPPB [13]. With these considerations, we evaluated ROS levels also in NB cells treated with BPPB, since this experiment would have been particularly helpful to understand whether (i) these nanovesicles’ preparation exerts its cytotoxic action on HTLA 230 and ER cells through ROS production, as in PLX-sensitive CMM cells; or (ii) the resistance acquired by HTLA ER based on high levels of antioxidants, following chronic treatment with etoposide [16], conferred resistance also to a preparation based on the use of nanovesicles (BPPBs). Moreover, the evaluation of ROS levels can be an indirect method to ascertain and confirm the presence of apoptosis, since as reported, low levels of ROS induce apoptosis and high levels induce necrosis [40,41].Therefore, to investigate the pro-oxidant action of BPPB also in HTLA 230 and HTLA ER NB cells, ROS levels were evaluated in both NB cell populations exposed for 24–72 h to increasing concentrations (0.1–2 µM) of BPPB using the DCFH essay [24] (Figure 4). The results were reported as a percentage of cells positive to DCFH normalized for protein content. Raw fluorescence data are, in fact, not applicable since an increase in fluorescence could be due to a high number of cells but also to an increased production of H_2_O_2_ induced by the treatments. This is the reason why it is necessary to normalize the fluorescence data to the protein content. Other cellular assays measuring oxidative damage, including lipid peroxidation or protein carbonylation, are likely too complex for low-tier early hazard assessment or hazard screening [42]. Conversely, the cellular DCFH assay is a simpler and widely used alternative, which measures intracellular ROS production by forming a fluorescent product (DCF) upon oxidation by free radicals produced by cells [43].

As shown in Figure 4, ROS production was moderately stimulated in both cell populations exposed to BPPB. In detail, a significant increase in ROS levels was observed in HTLA 230 cells treated with BPPB concentrations ≥ 0.5 µM at 24 (*p* = 0.0109) and 48 (*p* < 0.0001) h of exposure, and with concentrations ≥ 0.25 µM at 72 h. In fact, the highest values of DCFH-positive cells were 2.96 and 3.05, respectively, for 24 and 48 h treatments, with BPPB reaching the value of 5.8 when HTLA 230 was exposed to the highest doses for 72 h (Figure 4a). In HTLA ER cells, the highest values were 4.92 (vs. 2.96 found in HTLA 230) and 3.67 (vs. 3.05 found in HTLA 230) at 24 and 48 h of exposure, respectively (Figure 4b). On the contrary, after 72 h treatments, the maximal value (4.99) was 1.16 times lower than that detected in HTLA 230 cells, thus confirming the major tolerance of HTLA ER to the effects of BPPB for longer periods of treatments. However, for HTLA ER cells, the treatment with BPPB concentrations ≤ 1 (24 h), 1.25 (48 h), and 0.75 (72 h) µM did not induce significant changes in ROS levels (Figure 4b). Considering a study by Almutairi et al., BPPB was 9.4–23.4-fold more efficient in inducing ROS overproduction in both the cell line populations considered here than the TiO_2_-Ta nanoparticles (NPs) reported by the authors. While the authors observed a significant induction of ROS overproduction in human neuroblastoma (SH-SY5Y) cells at 10 µg/mL (11.7 µM) concentrations, after 24 h exposure, BPPB, after the same time, significantly induced ROS overproduction at only 0.5 µM (HTLA 230) and 1.25 µM (HTLA ER) concentrations, respectively [17]. Also, in a study by Zeng et al., it was reported that the nano-assembly CA-4S_2_@ES-Cu triggered a ROS increase of about 3–4.5 times in hepatocellular carcinoma (HCC) cells with respect to the control, after 24 h exposure at a concentration of 0.02 µM [44]. In this case, after 24 h exposure, BPPB also caused an increase in ROS of >3 times in HTLA 230 and >4 times in HTLA ER, with respect to the control, but at concentrations decidedly higher. These findings make foreseeing lower oxidative stress for the tissues in a future in vivo administration, while maintaining a marked cytotoxicity on tumor cells based on apoptosis. Also, Brauer et al. monitored the ROS overproduction in A-375 cancer cells upon the administration of 30 µM vemurafenib (Vem), a chemotherapeutic drug used against melanoma, 50 µM TRAM-34 (TRAM), a potassium channel inhibitor, and the combination of two drugs [45]. After 24 h exposure, the authors observed an increase with respect to the control of about 38%, 30%, and 5%, respectively [45]. In this regard, our results demonstrated that after the same time exposure, BPPB caused a similar ROS increase, but at a concentration remarkably lower of 2 µM.

#### 2.3.1. Correlation Between ROS Overproduction and BPPB Concentrations

To evaluate the potential correlation between ROS production and BPPB concentrations, it has been applied the method previously described for cell viability (%) vs. BPPB concentrations and recently reported [13]. The nonlinear regression models and all associated parameters were achieved using Microsoft Excel software 365. The results are shown in Appendix A. The investigation disclosed that, except for the data obtained in experiments carried out on HTLA 230 cells for 72 h, which were best described by a nonlinear regression model of a Power type, the nonlinear regressions which best fitted all other series of data were of a second-order polynomial type. Based on R^2^ values (0.9884, 0.9886, 0.9862, and 0.9807), a good correlation between ROS increase induced by BPPB and its concentrations was observed for HTLA ER cells at all exposure timings tested and for HTLA 230 cells, when exposed to BPPB for only 24 h. On the contrary, very low correlations existed between the data for experiments on HTLA 230 cells when they were treated with BPPB for 48 and 72 h. In general, except for these two latter cases, it is possible to claim a concentration-dependent mechanism for the ROS generation induced by BPPB.

#### 2.3.2. Correlation Between ROS Overproduction and Exposure Time

Appendix A reports the results from investigations carried out to assess the role of a time-dependent mechanism for the ROS overproduction induced by BPPB. These studies were performed following the method recently described [13]. The nonlinear regression models and all associated parameters were achieved using Microsoft Excel software 365. Based on R^2^ values (R^2^ = 1) (Appendix A), strong correlations of a second-order polynomial type were found between the average of ROS production and exposure times in both types of cells, thus confirming that ROS production was dependent on the time of treatments. Moreover, the results demonstrated that the average of ROS production in HTLA 230 cells was higher than that in HTLA ER by 1.3 (48 h) and 1.4 times (72 h), while the ROS levels’ increase induced by BPPB was similar in both cell populations when treated for 24 h. Moreover, while the average DCFH-positive cells (%) increased, passing from 24 to 48 and then to 72 h exposure in HTLA 230 cells, it decreased, passing from 24 to 48 h exposure, followed by a significant increase, passing from 48 to 72 h treatment in HTLA ER cells. In this regard, it could be hypothesized a temporary adaptation of these cells to the effects of BPPB over time, which, however, stopped working with longer treatments, determining a further increase in ROS.

#### 2.3.3. Correlation Between BPPB Cytotoxic Effects (% Cell Viability) and BPPB-Induced ROS Production

The results concerning the effects of BPPB on cell viability (Figure 1a,b) and those of ROS overproduction induced by BPPB (Figure 4a,b), may envisage the existence of correlations between the cytotoxic effects of BPPB and its ability to induce an increase in ROS amount and therefore a ROS-dependent anticancer mechanism. Therefore, by carrying out experiments like those performed in the previous sections for other series of data, this hypothesis was investigated. The results have been shown in Appendix A, which reports the dispersion graphs of the DCFH-positive cells (%) vs. cell viability (%) (round red and purple, square pink and blue, and triangular light pink and sky blue indicators without lines) observed at the same concentrations of BPPB, for both HTLA 230 and HTLA ER cells, respectively, after 24, 48, and 72 h of treatment. The best fitting nonlinear regression models (dotted lines), their equations, and the related R^2^ values were also included in Appendix A. All graphs and related parameters were provided by Microsoft Excel 365. Appendix A evidenced that the nonlinear regression models which best described the data of dispersion graphs were of a Power type for the experiments on HTLA 230 cells at 48 and 72 h of exposure, while of second-order polynomial types for the experiments on HTLA 230 cells at 24 h exposure and for those on HTLA ER at all exposure timings tested. Considering the experiments on the HTLA 230 cells, the R^2^ value of the nonlinear regression models was >0.95 (0.9738, 0.9529) when treated for 24 and 48 h, while it was <0.95 (0.8998) when cells were treated for longer periods (72 h). This finding confirmed that only when HTLA 230 cells were exposed to BPPB for 24 and 48 h, an inverse correlation of a second-order polynomial type and Power type, respectively, between ROS hyperproduction and cell viability (%) existed. Considering the experiments on HTLA ER cells, the R^2^ value of the nonlinear regression models was >0.95 (0.9779) when treated for 24 h, while it was <0.95 (0.8960, 0.9388) when cells were treated for longer periods (48 and 72 h). This finding confirmed that only when HTLA ER cells were exposed to BPPB for 24 an inverse correlation of a second-order polynomial type between ROS overproduction and cell viability (%) existed. Collectively, only for the cytotoxic effects of BPPB on HTLA 230 cells treated for 24 and 48 h, and on HTLA ER treated for 24 h, it was possible to claim a ROS-dependent mechanism. The dependence of the cytotoxic effects of BPPB on ROS overproduction in HTLA 230 cells treated for 72 h, and in HTLA ER cells treated for 48 and 72 h, was not conceivable. From these early experiments, the hypothesis of an apoptotic cell death induced by ROS can be considered.

### 2.4. Concentration- and Time-Dependent Cytotoxic Effects of BPPB on Astrocyte and Neuron Primary Cell Cultures

Similarly to the assessment of haemolytic toxicity, the evaluation of cytotoxicity towards mammalian cells is another pivotal step to afford in the early stage of drug development, before undertaking other costly and time-consuming in vivo experiments [46]. Indeed, it is necessary to predict if a new bioactive compound under investigation could be suitable for being developed as a novel agent for clinical application or could be exploited as a template molecule for the development of more active and less toxic derivatives by chemical transformations, or it is not advisable for further studies. The cytotoxicity of quaternary phosphonium salts (QPSs) depends mainly on their physicochemical characteristics, including the number of carbon atoms of alkyl chains, the presence of aromatic ring(s) on which the P^+^ cationic charge can delocalize [14,36], the type of counter anions, the possible colloidal properties, and the probable capability to form nano aggregates [47]. The specific types of cells exposed to QPSs can further influence their cytotoxicity [47]. In this regard, the constituents of membranes of cancer cells are different from those of mammalian cells [48]. This makes them more susceptible to the disruptive action of cationic compounds, thus making QPSs’ possible cytotoxicity towards tumour cells higher than towards normal ones [48]. Anyway, a certain cytotoxicity could manifest also in normal cells and quantifying it is mandatory [47]. Based on these considerations, although NB is a neuroendocrine tumour which derives from neural crest cells, we investigated the BPPB toxic effects on primary cell cultures, namely spinal cord astrocytes and cortical neurons. Astrocyte primary cultures have emerged as an important tool for understanding the responses of normal cells under both physiological and pathological conditions [49]. On the other hand, primary neuronal cell cultures are best suited for drug discovery and drug development. Primary cell cultures freshly prepared from animals reproduce physiological conditions astonishingly like the in vivo situation [50]. As for our knowledge, it is the first time that the cytotoxicity of a bola amphiphilic compound is assessed on primary cultures of astrocytes and neurons. More generically, the existent studies on the cytotoxicity of synthetic QPSs, or of a generic exogenous synthetic cationic organic compound or polymer on astrocytes and neurons primary cell cultures, are limited. As examples, upon a rapid survey, we have found a study on the mechanism by which amine-modified cationic polystyrene nanoparticles induced cell death in a human brain astrocytoma cell line [51], but not in primary astrocytes. A study on the mechanism by which endogenous eosinophil cationic protein/RNase 3 induced cell death in primary astrocytes was found [52], but exogenous synthetic cationic compounds were not considered. Concerning primary neurons, the cytotoxicity of both a commercial (LA2000) and a newly prepared cationic lipid (Arg (TFA^−^)) vs. primary neurons was assessed by Aoshima et al. [53]. The cytotoxicity of a library of 80 compounds, among which only 2/3 molecules were cationic salts, was evaluated in human-induced pluripotent stem cell (iPSC)-derived neural stem cells, neurons, and astrocytes by Pei et al. [54]. Vidal et al. assessed the neurotoxicity of a cationic fourth generation (G4) amine dendrimer (G4-NH_2_) towards primary neuron cell cultures. Significant cytotoxic effects and alterations in normal synaptic activity, which are generated by the enhanced membrane permeability and a subsequent intracellular Ca^2+^ increase, were observed after only 24 h exposure [55]. Also, Albertazzi et al. studied the cytotoxic effects of cationic G4-NH_2_ and C12 chain-modified G4-NH_2_ dendrimers towards primary cortical neurons, detecting the dramatic apoptotic cell death of neurons in vitro at 100 nM concentrations, for the C12-modified G4-NH_2_ compound [56]. Cell viability in neurons and astrocytes, treated with increasing concentrations of BPPB for 24, 48, and 72 h, was assessed using the MTT essay, which is a method endowed with simplicity, convenience, quantitative and high throughput, as well as reproducibility [57]. MTT is a well-established method for assessing cell viability, and its reliability is supported by standardized protocols in many areas of research [57].

Figure 5a,b show the results of our investigations as bar graphs.

Spinal cord astrocytes well tolerated BPPB for concentrations ≤ 2.93 µM (cell viability 74.95%) and 1.17 µM (cell viability 72.17 and 69.31%) when exposed to BPPB for 24 and 48–72 h, respectively. After 24 h of treatment, BPPB was toxic for concentrations > 5.86 µM (cell viability 47.14 and 20.22%), for concentrations = 11.73 µM after 48 (cell viability 45.18%), and for concentrations ≥ 8.80 µM after 72 h of exposure (cell viability 42.70 and 41.60%). The viability of cells decreased under 50% at BPPB concentration = 8.80, 11.73, and 8.80 µM after 24, 48, and 72 h treatment, respectively. The results evidenced that, in astrocytes, there is a gradual decrease in cell viability when exposed to increasing concentrations of BPPB, but the viability (%) of cells dropped below 40% only in the 24 h treatments and at the highest concentration tested, while for longer treatments of 48 and 72 h, it remained above 40% (45.18 and 41.67%, respectively), even at the highest concentration considered. A different trend, with a threshold concentration above which cells were completely killed, was observed when BPPB was administered to normal human keratinocytes (HaCaTs), as reported in our recent work [13]. Although the results of cytotoxicity experiments using TPP-based bola amphiphilic nanoparticles (NPs) on primary astrocytes had not yet been published until now, we can envisage that BPPB-induced cytotoxic effects could be mediated by a mechanism, like that reported by Bexiga and co-worker for cationic nanoparticles, when administered to a human brain astrocytoma cell line [51]. The authors demonstrated that cell death occurred by an apoptotic program accompanied by mitochondria damage, with the activation of caspases 3/7 and 9 and cleavage of poly (ADP-ribose) polymerase (PARP)-1 [51]. An apoptotic pathway, but in this case, via cell surface interactions determining an increase in the free cytosolic Ca^2+^ concentration, was reported in another study on the neurotoxicity of endogenous eosinophil cationic protein/RNase 3 [52]. To evidence the lower cytotoxicity of BPPB vs. astrocytes with respect to that observed against both HTLA 230 and HTLA ER cells, the graphs of their viability (%) vs. increasing concentrations of BPPB in the range of 0.1–1.75 µM are included in Figure 6.

Unequivocally, the difference in cell viability (%) is more evident if astrocytes and drug-sensitive HTLA 230 cells are considered. Indeed, the cell viability (%) of astrocytes was higher than that of HTLA 230 cells at all concentrations and all times considered. Anyway, even if for BPPB concentrations < 1 µM, the viability of MDR neuroblastoma cells was higher or like that of astrocytes, depending on exposure timing, for concentrations ≥ 1 µM, an inverse tendency was observed, and the viability (%) of astrocytes was higher than that of tumoral cells at all times considered. Cortical neurons were less tolerant than astrocytes to the cytotoxic effects of BPPB. Indeed, astrocytes have the function of preserving the brain microenvironment homeostasis to protect other brain cells, mainly neurons, against damage [58]. Regarding this, as in our experiments, the analysis of the cytotoxicity of 80 compounds towards four types of cell cultures, including primary astrocytes and neurons reported by Pei et al., indicated that astrocytes were less affected than the other three cell types [54]. In our context, the viability of cells was higher than 50% for concentrations ≤ 1.17, 1.80, and 0.29 µM (cell viability 54.64, 57.15, and 71.80%) when cells were exposed to BPPB for 24, 48, and 72 h, respectively. For higher concentrations, like in astrocytes, a gradual decrease in cells’ viability upon increasing concentrations of BPPB up to a concentration of 11.73 µM was observed, reaching cell viability percentages of 14.80, 20.03, and 10.63 after 24, 48, and 72 h treatment, respectively, at the highest concentration tested (11.73 µM, 10.0 µg/mL). The cytotoxicity of BPPB towards primary neuron cell cultures was anyway lower than that reported by Vidal et al. [55]. Specifically, the reduction in neuron viability by G4NH_2_ after 24 h exposure at 1 µM concentration was higher than that caused by BPPB at 1.17 µM (cell viability 44.57 vs. 54.63%), and the cytotoxic effects caused by G4NH_2_ at a 10 µM concentration were higher by 2.3 times (6.3 vs. 14.8%) than those caused by BPPB at the highest concentration tested of 11.73 µM. Furthermore, the cytotoxicity of BPPB towards primary neuron cell cultures was significantly lower than that reported by Albertazzi et al. for a cationic C12-modified G4NH_2_ dendrimer [56]. Specifically, dramatic apoptotic cell death of neurons was observed in vitro by the authors at 0.1 µM concentration; the viability of primary cortical neurons exposed to 0.12 µM of BPPB was close to 90% (89.49%) after 24 h exposure, and even higher than 90% after 48 and 72 h treatments (96.77 and 95.90%, respectively). Note that, in both astrocytes and neurons, a sort of adaptation of cells to BPPB effects with an increase in their tolerance occurred for treatments longer than 24 h. At this point, we considered that, as reported in Table 2, the IC_50_ values of BPPB on primary cultures of cortical astrocytes and neuronal cells were approximately 7.044 and 1.56 µM, respectively, while apoptotic death in HTLA-230 cells and in HTLA ER cells was recorded in cells treated with 0.5 and 1 µM, respectively. Therefore, since the compound given at the concentrations capable of causing approximately 50% of cell death and an overproduction of ROS in HTLA-230 and in HTL ER cells was not cytotoxic for astrocytes and neurons, evaluating apoptosis and ROS in normal cells treated with these concentrations lower than the IC_50_ of the primary lines for comparison purposes would not have led to any relevant results and would only have resulted in a waste of time and unnecessary additional expenses. Following the same procedure reported for NB cells in Section 3.2.1, the IC_50_ values of BPPB vs. both primary cell cultures were calculated. Figure 7 and Figure 8 show the dispersion graphs of cell viability (%) vs. BPPB concentrations concerning spinal cord astrocytes (Figure 6a) and cortical neurons (Figure 6b) and those of cell viability (%) vs. Log BPPB concentrations with the nonlinear regression models vs. the normalized cell response (Figure 7a,b), respectively. Table 2 reports the calculated IC_50_ values.

### 2.5. Selectivity Index

The selectivity index (SI) of BPPB for NB cells with respect to RBCs, as well as cortical neuron and astrocyte primary cells, was calculated to predict its possible clinical use and therapeutic potential, according to equation (Equation (1)), as follows:SI = IC_50_ for NTC/IC_50_ for NB cells(1)
where NTC means not tumoral cells. The value of SI is a cardinal check to establish if a new molecule could be worthy of consideration for further studies and future development as a new therapeutic agent. Although opinions are contrasting, according to a recent article by Krzywik et al., a favourable SI > 1.0 indicates a drug with efficacy against tumour cells greater than the toxicity against normal cells and could establish a high potential for clinical development [59]. On the contrary, Peña-Morán et al. reported that according to the “selectivity criteria”, only compounds with SI > 10 could be considered selective for a certain cell line, while compounds with SIs lower than 10 but higher than 1 could be considered non-selective [60].

The IC_50_ values determined on HTLA 230 and HTLA ER cells are included in Table 3, together with the HC_50_ determined on RBCs and the IC_50_ values determined on primary cortical astrocytes and neurons for a direct comparison.

The IC_50_ values reported in Table 3 were determined after 24, 48, and 72 h of exposure of NB cells and primary cortical astrocytes and neurons to BPPB, and the HC_50_ determined on RBCs after the time of experiments, as for protocol [61], was used to calculate the SIs according to Equation (1), as reported in Table 4.

SI values higher than 10 and up to 39.98 were determined for both NB cell populations with respect to RBCs and at all exposure timings tested, thus establishing that concerning its possible haemolytic toxicity, BPPB possesses the requirements to be further studied as an anticancer therapeutic approach for clinical development, also according to the strict “selectivity criteria” reported by Peña-Morán et al. [60]. High SI values >>> 1 and in the range of 4.25–23.73 were determined for HTLA 230 and HTLA ER cells with respect to primary spinal cord astrocytes at all exposure timings, while SI values > 1 and in the range of 1.26–4.78 were determined for HTLA 230 and HTLA ER cells with respect to primary cortical neurons, at all exposure timings, except for HTLA ER when neurons were treated for long periods (0.64). The trend of SIs as functions of exposure timing has been shown in Figure 9, reporting in the graph the SI values of BPPB for HTLA 230 and HTLA ER cells concerning its haemolytic toxicity (Figure 9a) and cytotoxicity vs. normal astrocytes and neurons (Figure 9b) as the function of time of exposure.

As for RBCs, while the SI values determined for the HTLA 230 cells increased by 2 times, passing from treatments of 24 h to treatments of 48 h, and by a further 1.2 times, passing from treatments of 48 h to 72 h, those determined for the HTLA ER cells remained practically constant. The SIs for both NB cells were high when astrocytes were considered, especially for the intermediate exposure timing of 48 h, while SI values were lower for neurons, thus confirming that neurons are more susceptible to the cytotoxic effects of BPPB. The higher susceptibility of neurons to molecules such as BPPB, which are reported to act by impairing membranes and mitochondria functions [36], may probably depend on the higher resistance of their cellular and mitochondria membranes with respect to those of neurons. This characteristic has been documented by Gürer et al., who demonstrated that astrocytes are more resistant to injuries than neurons [62]. In their study, they found that, after 2 h middle cerebral artery occlusion in mice, astrocytes, in contrast to neighbouring neurons, were alive, contained glycogen across the ischemic area 6 h after reperfusion, had intact plasma membranes, and maintained mitochondrial integrity [62]. Furthermore, the authors demonstrated that astrocyte ultrastructure, including mitochondria and lysosomes, disintegrated much later than that of neurons, and they died by a delayed necrosis without significantly activating apoptotic mechanisms [62]. Anyway, also for neurons, higher SI values were observed when cells were treated for 48 h, as in the case of astrocytes. Collectively, it seems like a sort of transient adaptation can occur in both cell cultures over time, making cells temporarily more tolerant to BPPB.

### 2.6. Correlation Between BPPB Cytotoxic Effects and BPPB Concentrations or Exposure Time on Astrocytes and Neuron Primary Cell Cultures

The dispersion graphs of the cell viability (%) of both primary spinal cord astrocytes and cortical neurons after 24, 48, and 72 h of exposure to increasing BPPB vs. BPPB concentrations of 1.1–11.7 µM (1–10 µg/mL), and their best fitting nonlinear regression models, were used to establish if the cytotoxic effects of BPPB are mediated by a concentration-dependent mechanism. Following the method recently reported [13] and previously used for the results on NB cells, we investigated the correlation between BPPB anticancer effects and its administered concentrations, based on the R^2^ values of the statistical models which best fitted the data of dispersion graphs. The nonlinear regression models and all associated parameters were achieved using Microsoft Excel software 365, as shown in Appendix A. According to the R^2^ values, which were all >0.95, a concentration-dependent mechanism can be assumed for the cytotoxic effects of BPPB to both cell cultures, and especially for those to neurons and at all exposure timings. Especially, acceptable to good correlations of a second-order polynomial (24 h) and logarithmic (48 and 72 h) type between cell viability (%) and BPPB concentrations were detected for astrocytes, while strong logarithmic (24 and 48 h) and Power (72 h) correlations were observed for neurons. Similarly, the possible existence of a time-dependent mechanism for the cytotoxic effects of BPPB (Appendix A) was investigated. According to R^2^ values, which were both =1, a very strong correlation between the cytotoxic effects of BPPB and exposure timing exists for both cell cultures, thus unequivocally establishing the existence of a time-dependent mechanism for the cytotoxic effects of BPPB. These findings evidence a substantial difference between the modalities by which BPPB acts on cancer cells, where concentration-dependent mechanisms cannot always be assumed, with respect to normal cells.

## 3. Materials and Methods

### 3.1. Chemicals and Instruments

All reagents and solvents used in this study were obtained by Merk (Milan, Italy) and were used without further purification. The compound named 1,1-(1,12-dodecanediyl)-*bis*-[1,1,1]-triphenyl phosphonium di-bromide (BPPB) was synthetized and characterized, as recently described [14].

### 3.2. In Vitro BPPB Cytotoxicity Evaluation on NB Cells

#### 3.2.1. Cells and Culture Conditions

HTLA 230 is a MYCN-amplified human stage-IV NB cell line, obtained from the G. Gaslini Institute, Genoa, Italy [63]. The cytogenetic features of the HTLA 230 cell line include 4p MYCN amplification, del(11)t(11;Y), balanced translocation t(1;17)(p36;q21), and dup(11p), which were previously described by Pezzolo et al. [64]. Cells were periodically tested for mycoplasma contamination (Mycoplasma Reagent Set, Aurogene s.p.a, Pavia, Italy). After thawing and eight passages in the culture, cell morphology and proliferation were analyzed. The MDR cell line (HTLA ER) was selected by treating HTLA 230 cells for 6 months, with increasing concentrations of etoposide (Calbiochem, Merck KGaA, Darmstadt, Germany) up to 1.25 μM, and then maintaining them in a medium supplemented with 1.25 μM etoposide, a dose comparable to that clinically used [16,65]. Both cell populations were cultured in RPMI 1640 (Euroclone SpA, Pavia, Italy) supplemented with 10% fetal bovine serum (FBS; Euroclone, Pavia, Italy), 2 mM glutamine (Euroclone, Pavia, Italy), 1% penicillin/streptomycin (Euroclone, Pavia, Italy), 1% sodium pyruvate (Sigma-Aldrich, Sant Louis, MO, USA), and 1% amino acid solution (Sigma) [63].

#### 3.2.2. Treatments

To determine the cytotoxic effects of BPPB, in vitro time- and dose-dependent experiments were carried out, as recently described [11]. Briefly, cells were treated for 24, 48, and 72 h, with increasing concentrations (0.1–2.0 µM) of the compound. The stock solution of BPPB was prepared in 40,000-fold diluted DMSO, and pilot experiments demonstrated that the final DMSO concentrations did not change any of the analysed cell responses. Cell cultures were carefully monitored before and during the experiments to ensure optimal cell density. Notably, samples were discarded if the cell confluence reached >90%.

#### 3.2.3. Cell Viability Assay

Cell viability was determined by using the CellTiter 96^®^ AQueous One Solution Cell Proliferation MTS Assay (Promega, Madison, WI, USA), as previously described [11,37,66]. MTS is a colorimetric method commonly used for determining cell viability. The cell survival rate, expressed as the cell viability percentage (%), was evaluated based on the experimental outputs of treated cells vs. the untreated ones (CTRs), and was calculated as follows: cell viability (%) = (OD treated cells − OD blank)/(OD untreated cells − OD blank) × 100%. IC_50_ values were calculated by GraphPad Prism 8.0.1 Software (GraphPad Software v8.0, San Diego, CA, USA), as detailed in the Results and Discussion in Section 2.

#### 3.2.4. DAPI Staining

After the treatment, cells were washed with PBS and stained with the fluorescent DNA dye DAPI (1 µg/mL) for 30 min. After additional washing, the images were immediately acquired in the fluorescence mode using a Nikon AX R confocal microscope equipped with a PLAN APO 20× (Nikon Europe, B.V., Amstelveen, The Netherlands).

#### 3.2.5. Annexin-V and Propidium Iodide Staining

Apoptotic and necrotic cells were analysed, as described in Marengo et al. [24]. After the treatment, NB cells were incubated with fluorescein isothiocyanate (FITC)-labelled recombinant Annexin-V and propidium iodide (PI; Biotium, Fremont, CA, USA), following the manufacturer’s protocol. After staining, the images were immediately acquired in fluorescence mode using a Nikon AX R confocal microscope equipped with a PLAN APO 20× (Nikon Europe, B.V., Amstelveen, The Netherlands).

### 3.3. Detection of Hydrogen Peroxide (H_2_O_2_) Production

The production of H_2_O_2_ was evaluated using 2′-7′-dichlorofluorescein-diacetate (DCFH-DA; Merk Life Science S.r.l. Milan, Italy), as previously reported [16,37,66]. Briefly, NB cells (10^4^ cells/well) were seeded in 96-well plates (Corning) and treated. Then, cells were stained with 2′-7′ di-chloro-fluorescein-diacetate (DCFH-DA; Sigma-Aldrich) and incubated with 90% DMSO for an additional 10 min in the dark. The generated fluorescence intensity was monitored with a Perkin Elmer fluorometer (Perkin Elmer Life and Analytical Sciences, Shelton, WA, USA) at 485/530 nm excitation/emission. Values were normalized to the protein content.

### 3.4. In Vitro Dose- and Time-Dependent Cytotoxicity of BPPB Against Spinal Cord Astrocytes and Cortical Neurons’ Primary Cell Cultures

#### 3.4.1. Spinal Cord Astrocyte Primary Cell Cultures

Astrocytes were prepared from the spinal cord of neonatal 2-day-old C57BL/6J mice as previously described, with minor modifications [67]. Briefly, two days after birth, pups (P1-2) were euthanized by cervical dislocation by trained personnel, and spinal cords were rapidly removed. Each dissected spinal cord was gently homogenized in 1 mL Dulbecco’s modified Eagle medium (DMEM; Euroclone, Pavia, Italy, Cat# ECM0728L) containing 10% fetal bovine serum (FBS, Euroclone, Pavia, Italy, Cat# ECS0180L), 1% glutamine (Euroclone, Pavia, Italy, Cat# ECB3004D), and 1% penicillin/streptomycin (Euroclone, Pavia, Italy, Cat# ECB3001D). Then, tissue suspension was seeded in a 35 mm Petri dish (Euroclone, Pavia, Italy, Cat# ET2035), pre-coated with poly-L-ornithine hydrochloride (PO, 1.5 µg/mL; Sigma, Milan, Italy, Cat# P2533). The preparations were maintained at 37 °C in a humidified 5% CO_2_ incubator for 5 days; then, the medium was replaced with fresh complete DMEM. After 7 days in vitro (DIV), cells were detached using Trypsin-EDTA 1X (Euroclone, Pavia, Italy, Cat# ECB3052B) and replated into a pre-coated 35 mm Petri dish until confluence. The culture medium was systematically replaced with fresh medium every two days. Spinal cord primary cell cultures have been characterized as previously described, before their use in in vitro assays [67,68,69]. At 16 DIV, P2 astrocytes were detached and seeded at the optimal density of 20,000 cells/well in a PO pre-coated 96-well. The day after, astrocytes were treated with bis-phosphonium bromide (BPPB), as previously described. The pups used for the spinal cord astrocyte primary cell culture preparations were utilized according to the European Communities Council Directive (EU Directive 114 2010/63/EU for animal experiments; 22 September 2010), and with the Italian D.L. No. 26/2014, and were approved by the local ethics committee and by the Italian Ministry of Health (Project Authorization No. 1022/2020-PR). The purity of neonatal astrocytes was confirmed by flow cytometry, labeling cell suspension with antibodies for GFAP or ACSA2, specific astrocyte markers. Astrocyte preparations expressed both the markers GFAP and ACSA2 (95.03 ± 3.38% GFAP-positive cells, 94.46 ± 1.92% ACSA2-positive cells, and GFAP and ACSA2 co-expressing positive cells), when compared to the respective unstained controls [67]. Moreover, cell suspensions showed a very low contamination of microglia cells, labeled with TMEM119 (2.42 ± 0.69% TMEM119-positive cells), compared to the respective unstained control. We also performed confocal microscopy studies staining spinal cord neonatal astrocytes with antibodies for GFAP and integrin alpha-M/beta-2 (CD11b; specific microglia marker). Astrocytes were efficiently stained with GFAP (green fluorescence), while they did not show contamination by microglia cells, labelled with CD11b (red fluorescence) [67]. Overall, these results indicated that the neonatal astrocyte primary cell culture preparations were pure astrocyte, not contaminated by microglia cells.

#### 3.4.2. Cortical Neuron Primary Cell Cultures

Cortical neurons were prepared from the cortex of E16-E17 Sprague Dawley rat embryos, as previously described [70]. The cortex was removed under binocular dissection (Nikon SMZ-2T, Tokyo, Japan) in a physiologic medium composed of 120 mM NaCl, 5 mM KCl, 25 mM HEPES, and 9 mM glucose (HIB, pH 7.4) at 4 °C. Tissue was chopped with a sterile blade, transferred into a 50 mL tube, and digested in 0.5% trypsin from bovine pancreas (Sigma, Milan, Italy, Cat#T1426-50MG) and 40 mg Deoxyribonuclease I from bovine pancreas (Sigma, Milan, Italy, Cat#DN25-100MG) for 10 min at 37 °C. Trypsin digestion was stopped by adding 1.5 mL FBS (Euroclone, Pavia, Italy, Cat# ECS0180L). The preparation was then centrifuged at 3300 rpm for 1 min and the pellet was resuspended in a plating medium composed of 10 mL neurobasal medium (Thermo Fisher Scientific, Monza, Italy, Cat# 21103-049), 10% FBS (Euroclone, Pavia, Italy, Cat# ECS0180L), 1% glutamine (Euroclone, Pavia, Italy, Cat# ECB3004D), and 1% penicillin/streptomycin (Euroclone, Pavia, Italy, Cat# ECB3001D). Tissue was then triturated 4-5 times with 10 mL pipette and filtered through 100 µm plastic mesh (Greiner BIO-ONE, Cassina de Pecchi, Italy, Cat#542000). Subsequently, 50,000 cells were seeded into 96-well plates previously coated with 1.5 µg/mL poly-L-ornithine hydrochloride (Merck, Milan, Italy, Cat# P2533) and 3 µg/mL laminin (Merck, Milan, Italy, Cat# L2020). After 60 min, the plating medium was replaced by a maintenance medium composed of neurobasal medium, 1% B27 (ThermoFisher Scientific, Monza, Italy, Cat# 17504044), 0.5% penicillin/streptomycin (Euroclone, Pavia, Italy, Cat# ECB3001D), and 0.5% glutamine to reduce astroglial cell contamination. The medium was systematically replaced with fresh medium every two days. At 14 DIV, cortical neurons reached the correct maturation and were treated with BPPB, as previously described. The rat embryos used for spinal cord cortical neuron primary cell culture preparations were utilized according to the European Communities Council Directive (EU Directive 114 2010/63/EU for animal experiments; 22 September 2010) and with the Italian D.L. No. 26/2014 and were approved by the local ethics committee and by the Italian Ministry of Health (Project Authorization No. 2018-75f11.N.POG, 2023-75F11.N.FIE). Cortical neurons’ identification was made by incubating cells with specific primary antibodies including Tau (axon microtubule-associated protein, mouse monoclonal 1:500, Synaptic System, Göttingen, Germany), MAP2 (dendritic microtubule-associated protein, rabbit polyclonal 1:500, Synaptic System), and DAPI (nuclei, 1:1000, Synaptic System) [70]. Additionally, cultures were exposed to secondary antibodies, such as Alexa Fluor 488 (1:700, Invitrogen, Waltham, MA, USA) and Alexa Fluor 549 (1:1000, Invitrogen), Goat anti-mouse, or Goat anti-rabbit. Images were acquired with a fluorescence-equipped microscope (Olympus BX-51) by using a CCD camera Orca ER II, Hamamatsu (Digital Imaging Systems Ltd, Egham, UK) and Image ProPlus software 7.0 (Media Cybernetic, Digital Imaging Systems Ltd, Egham, UK) [70]. Cells expressed all specific neuron markers.

#### 3.4.3. MTT Cytotoxicity Assay

To assess the cytotoxic properties of BPPB, the MTT (3-(4,5-dimethylthiazol-2-yl)-2,5- diphenyltetrazolium bromide) assay (Abcam, Milan, Italy, Cat#ab2011091) was performed, following the manufacturer’s protocol, as previously reported [11,14,71]. Spinal cord astrocytes and cortical neurons were treated with BPPB, diluted in complete DMEM, at increasing concentrations (range 0.1 µg/mL-10 µg/mL) and incubated at 37 °C in 5% CO_2_ for a further 24, 48, and 72 h. After that, the media were removed, and the cells were washed with PBS. Aliquots (200 µL) of serum-free medium containing MTT (Merck, Milan, Italy, Cat #M5655; 0.25 mg/mL MTT) were added to each well and incubated at 37 °C for 3 h. After removing the medium, 200 µL of DMSO solution (Merck, Milan, Italy, Cat #276855; 0.25 mg/mL MTT) was added to each well and horizontally shaken for 10 min to allow DMSO to solubilize the formazan crystals, allowing the formation of a homogenous solution. The 570 nm wavelength light absorption was then measured spectrophotometrically in each well using a TECAN instrument (TECAN ITALIA s.r.l., Milan, Italy) and converted into OD (optical density) units. The cell survival rate, expressed as a cell viability percentage (%), was evaluated based on the experimental outputs of the treated groups vs. the untreated groups (CTRs), and was calculated as follows: cell viability (%) = (OD treated cells − OD blank)/(OD untreated cells − OD blank) × 100%.

### 3.5. Statistical Analyses

The results have been expressed as the means ± S.D. of at least four independent experiments, in which six different wells were analysed every time for each experimental condition. In the analysis of cell viability or H_2_O_2_ levels, the condition of untreated cells (Ctr) was set as 100% ± S.D. and 1 ± S.D., respectively. The statistical significance of differences was determined by ordinary one-way analysis of variances (ANOVAs) followed by Dunnet’s multiple comparison test correction using GraphPad Prism 8.0.1 (GraphPad Software v8.0, San Diego, CA, USA). The asterisks indicate the following *p*-value ranges: * *p* < 0.05, ** *p* < 0.01, *** *p* < 0.001, **** *p* < 0.0001. *p* > 0.05 was not considered statistically significant and no symbol was used in the images.

## 4. Conclusions

In this study, the effects of synthesized TPP-based bola amphiphilic nanovesicles (BPPBs) on cell viability and ROS production were assessed towards drug-sensitive (HTLA 230) and MDR (HTLA ER) NB cells. The mechanisms governing the cytotoxic effects of BPPB were explored both by correlation studies and experiments to evaluate apoptosis and necrosis. By this approach, we have demonstrated that BPPB exerts its cytotoxicity in a time-dependent mode and that a moderate increase in ROS levels triggered mainly apoptotic death without necrosis, caused instead by high levels of ROS, thus avoiding the emergence of secondary inflammation. Furthermore, in this study, for the first time, the in vitro cytotoxicity of BPPB bola amphiphilic nanovesicles was assessed on primary spinal cord astrocyte and cortical neuron culture cells, thus evidencing a substantial difference between the modality of BPPB action on cancer cells in respect to healthy cells These last findings associated to the low haemolytic activity of BPPB previously assessed, the apoptotic mode of action and the excellent data of cytotoxicity which establish that BPPB is more efficacious than other nanoparticles previously described, pave the way for future investigations in in vivo models. Indeed, further improvements in the anticancer effect of BPPB by reducing its residual cytotoxicity on not tumoral cells could derive from chemical modifications, such as varying the length of the alkyl chain acting as a linker between the two cationic heads. Furthermore, its formulation in cationic liposomes could render BPPB a novel potent device or a template molecule to be developed as a new clinically applicable agent for counteracting MDR NB.

## Figures and Tables

**Figure 1 ijms-26-04991-f001:**
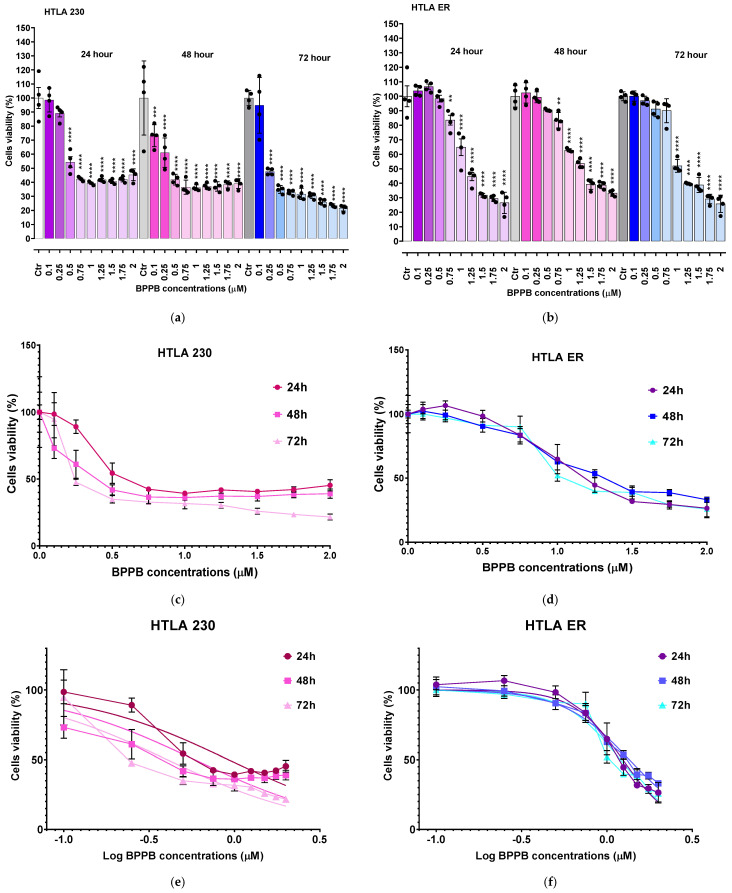
Cell viability was evaluated in HTLA 230 (**a**) and in HTLA ER (**b**) cells exposed to increasing concentrations of BPPB (0.1–2.0 µM) for 24, 48, and 72 h. Bar graphs summarize quantitative data of the means ± S.D. of four independent experiments (n = 4, round black spheres) run in triplicates. Significance is versus control condition (*). Specifically, **** *p* < 0.0001; *** *p* < 0.001; ** *p* < 0.01 (one-way ANOVA followed by Dunnet’s multi-comparisons test). Dispersion graphs of cell viability of HTLA 230 (**c**) and HTLA ER (**d**) cells vs. increasing BPPB concentrations (0.1–2.0 µM) after 24 h (red line for HTLA 230 and purple line for HTLA ER), 48 h (pink line for HTLA 230 and blue line for HTLA ER), and 72 h (light pink line for HTLA 230 and sky-blue line for HTLA ER) of exposure. Concentration = 0.0 µM corresponded to the control. Plot of Log concentration of BPPB vs. cell viability of HTLA 230 (**e**) and HTLA ER (**f**) after 24, 48, and 72 h of exposure (red, pink, and light pink traces with indicators and error bars for HTLA 230; purple, blue, and sky blue traces with indicators and error bars for HTLA ER), and plot of nonlinear regressions of Log concentrations of BPPB vs. normalized response after 24, 48, and 72 h of exposure (red, pink, and light pink traces without indicators for HTLA 230 and purple, blue, and sky blue traces without indicators for HTLA ER).

**Figure 2 ijms-26-04991-f002:**
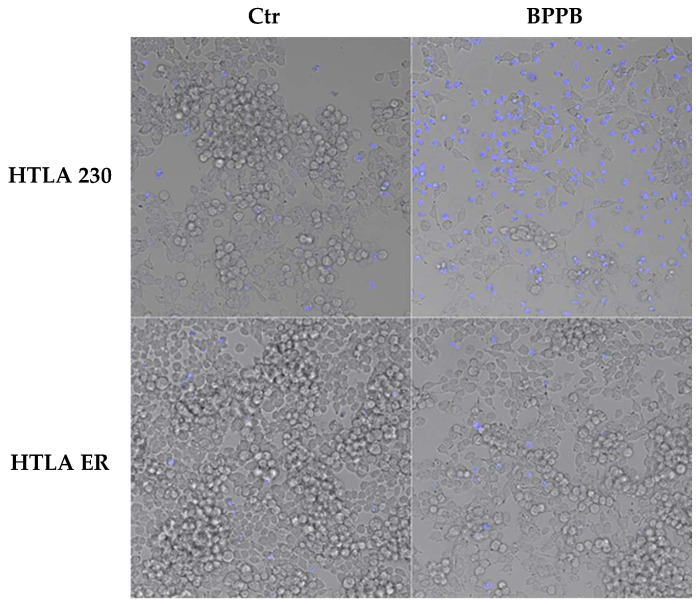
BPPB-induced necrosis of HTLA 230 cells. HTLA 230 and HTLA ER cells were treated for 24 h with 0.5 µM and 1 µM BPPB, respectively, and then labelled with DAPI. As shown in the fluorescent microscope merged images, the dead cells were labelled in blue, and each image was representative of three independent experiments. HTLA 230 = sensitive neuroblastoma cells; HTLA ER = multidrug-resistant neuroblastoma cells; BPPB = bis-phenyl phosphonium bromide; Ctr = control.

**Figure 3 ijms-26-04991-f003:**
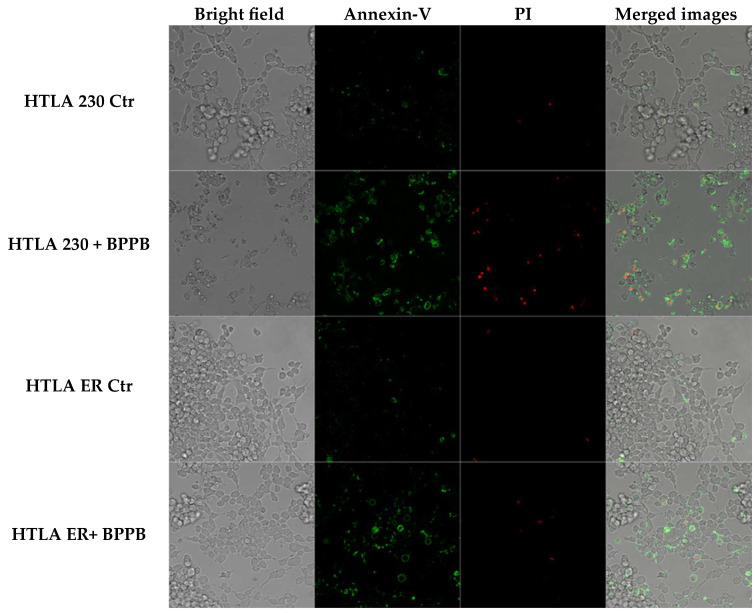
BPPB-induced apoptosis of HTLA 230 and HTLA ER cells. HTLA 230 and HTLA ER cells were treated for 24 h with 0.5 µM and 1 µM BPPB, respectively, and then labelled with Annexin-V and propidium iodide (PI). The first column of the panels shows images of cells observed by standard filters. The second column shows images of Annexin V-positive cells (apoptotic cells). The third column shows images of PI-positive cells (necrotic or late apoptotic cells). The fourth column shows the merged images. The reported images were representative of three independent experiments. HTLA 230 = sensitive neuroblastoma cells; HTLA ER = multidrug-resistant neuroblastoma cells; BPPB = bis-phenyl phosphonium bromide; Ctr = control.

**Figure 4 ijms-26-04991-f004:**
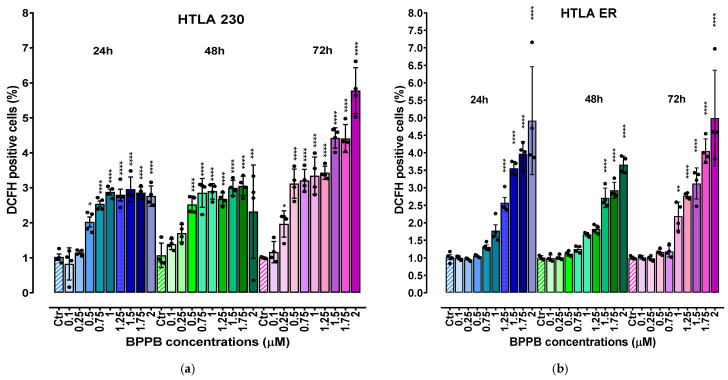
ROS production expressed as DCFH-positive cells was analysed in HTLA 230 (**a**) and HTLA ER (**b**) cells exposed to increasing concentrations of BPPB (0.1–2 µM) at 24 h (blue bars), 48 h (green bars), and 72 h (pink to purple bars). Bar graphs summarize quantitative data of the means ± S.D. of four independent experiments (n = 4, round black spheres) run in triplicates. Results were reported as percentage of cells positive to DCFH normalized for protein content. This value for the control (Ctr) was fixed to 1. Significance refers to control (*). Specifically, 0.01 < *p* < 0.05 *, *p* < 0.01 **, *p* < 0.001 ***, and *p* < 0.0001 **** (one-way ANOVA followed by Dunnet’s multi-comparisons test). HTLA 230 = sensitive neuroblastoma cells; HTLA ER = multidrug-resistant neuroblastoma cells; BPPB = bis-phenyl phosphonium bromide; Ctr = control; DCFH = 2′-7′-dichlorodihydrofluorescein.

**Figure 5 ijms-26-04991-f005:**
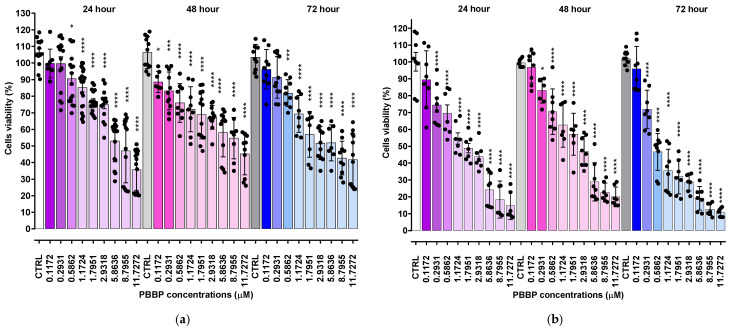
Cell viability was evaluated in spinal cord astrocytes (**a**) and in cortical neurons (**b**) exposed to increasing concentrations of BPPB (0.11–11.73 µM, i.e., 0.1–10.0 µg/mL) for 24, 48, and 72 h. Bar graphs summarize quantitative data of the means ± S.D. of n = 6–20 independent experiments (round black spheres) run in triplicate. Significance refers versus control (CTRL) conditions (*). Specifically, * 0.01 < *p* < 0.05, *** *p* < 0.001 **** *p* < 0.0001 (one-way ANOVA followed by Dunnet’s multi-comparisons test).

**Figure 6 ijms-26-04991-f006:**
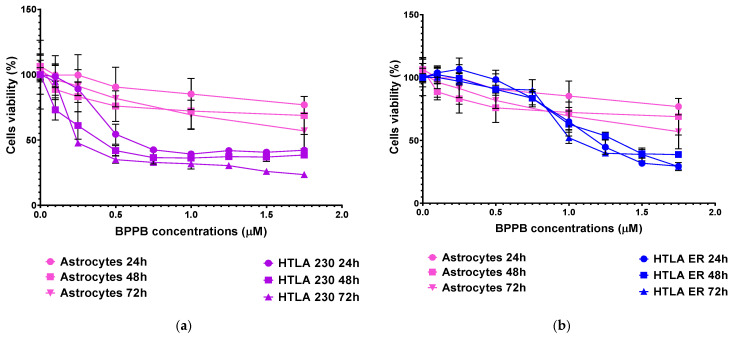
HTLA 230 (**a**), HTLA ER (**b**), and spinal cord astrocytes’ (**a,b**) viability (%) vs. increasing concentrations of BPPB in the range 0.1–1.75 µM.

**Figure 7 ijms-26-04991-f007:**
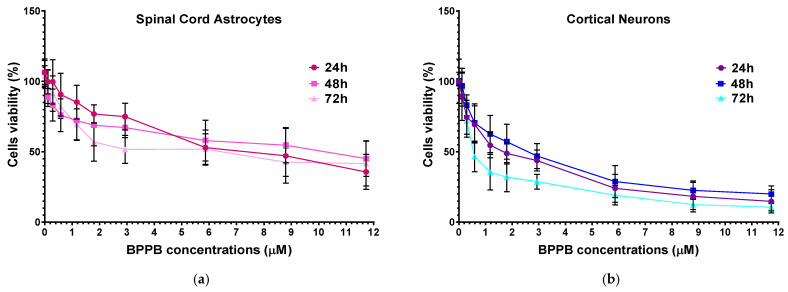
(**a**) Cell viability (%) of spinal cord astrocytes vs. increasing BPPB concentrations (0.11–11.73 µM) after 24 h (red line), 48 h (pink line), and 72 h (light pink line) of exposure. (**b**) Cell viability (%) of cortical neurons vs. increasing BPPB concentrations (0.11–11.73 µM) after 24 h (purple line), 48 h (blue line), and 72 h (sky blue line) of exposure. Concentration = 0.0 µM corresponded to the control.

**Figure 8 ijms-26-04991-f008:**
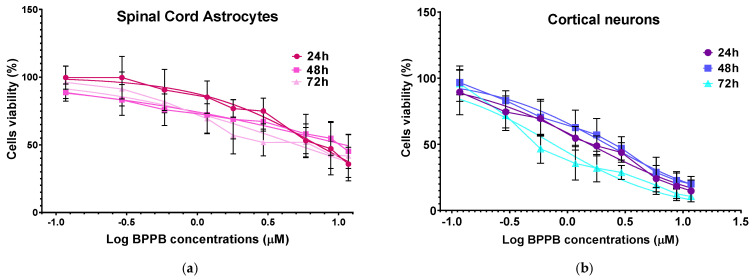
(**a**) Plot of Log concentration of BPPB vs. cell viability (%) of spinal cord astrocytes after 24, 48, and 72 h of exposure (red, pink, and light pink traces with indicators and error bars) and plot of nonlinear regressions of Log concentrations of BPPB vs. normalized response (Hill slope) after 24, 48, and 72 h of exposure (red, pink, and light pink traces without indicators). (**b**) Plot of Log concentration of BPPB vs. cell viability (%) of cortical neurons after 24, 48, and 72 h of exposure (purple, blue, and sky-blue traces with indicators and error bars) and plot of nonlinear regressions of Log concentrations of BPPB vs. normalized response (Hill slope) after 24, 48, and 72 h of exposure (purple, blue, and sky-blue traces without indicators).

**Figure 9 ijms-26-04991-f009:**
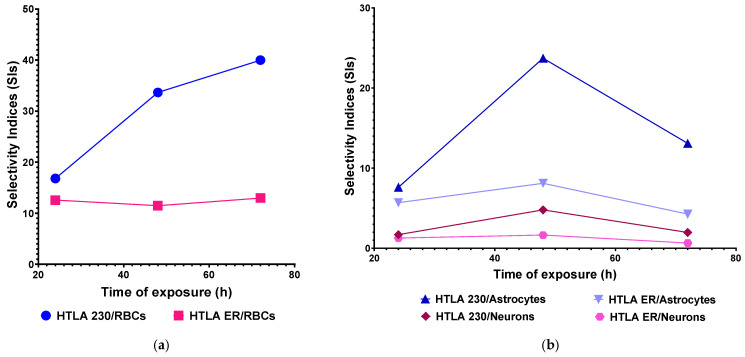
(**a**) SI values of BPPB for both HTLA 230 and HTLA ER cells in relation to its haemolytic toxicity as function of time of exposure. (**b**) SI values of BPPB for both HTLA 230 and HTLA ER cells in relation to its cytotoxicity vs. cortical astrocytes and neurons as function of time of exposure. RBCs = red blood cells.

**Table 1 ijms-26-04991-t001:** IC_50_ values of BPPB on HTLA 230 and HTLA ER NB cells measured from results of cytotoxicity carried out in the range of concentrations 0.1–2 µM.

Exposure Time (h)	IC_50_ HTLA 230 (µM)	IC_50_ HTLA ER (µM)
24	0.9257 ± 0.1637	1.2390 ± 0.0610
48	0.4623 ± 0.1031	1.3540 ± 0.0520
72	0.3892 ± 0.0836	1.1990 ± 0.0695

**Table 2 ijms-26-04991-t002:** IC_50_ values of BPPB on primary cortical astrocytes and neuron cell cultures measured from results of cytotoxicity carried out in the range of concentrations 0.12–11.73 µM (0.1–10.0 µg/mL).

Exposure Time (h)	Primary Spinal Cord Astrocytes (µM)	Primary Cortical Neurons (µM)
24	7.044 ± 0.4760	1.5610 ± 0.2775
48	10.9700 ± 5.0935	2.2080 ± 0.3365
72	5.0950 ± 1.2240	0.7631 ± 0.1270

**Table 3 ijms-26-04991-t003:** BPPB IC_50_ values vs. RBCs, cortical astrocytes, and neurons and NB cell populations.

IC_50_ (µM)	IC_50_ 24 h (µM)	IC_50_ 48 h (µM)	IC_50_ 72 h (µM)	HC_50_ Experiment Time (µM)
RBCs *	N.A.Q.	N.A.Q.	N.A.Q.	15.56 ± 12.13
Spinal cord astrocytes	7.044 ± 0.9805	10.9700 ± 5.0935	5.0950 ± 1.2240	----
Cortical neurons	1.5610 ± 0.2775	2.2080 ± 0.3365	0.7631 ± 0.1270	----
HTLA 230	0.9257 ± 0.1637	0.4623 ± 0.1031	0.3892 ± 0.0836	----
HTLA ER	1.2390 ± 0.0610	1.3540 ± 0.0520	1.1990 ± 0.0695	----

N.A.Q. = not acquired. * [13].

**Table 4 ijms-26-04991-t004:** Selectivity of BPPB for NB cells with respect to primary astrocytes, primary neurons, and RBCs expressed as selectivity index (SI).

Cells	SI 24 h ^a^	SI 48 h ^a^	SI 72 h ^a^
RBCs *	16.81	33.66	39.98
RBCs **	12.56	11.49	12.98
Spinal cord astrocytes *	7.61	23.73	13.09
Spinal cord astrocytes **	5.69	8.10	4.25
Cortical Neurons *	1.69	4.78	1.96
Cortical Neurons **	1.26	1.63	0.64

**^a^** Time of exposure of NB cells; * vs. HTLA 230 [13]; ** vs. HTLA ER [13].

## Data Availability

All data and information useful to readers to understand and correctly interpret this study are available in this article and related Appendix A.

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
