# Peer review of "TPP-Based Nanovesicles Kill MDR Neuroblastoma Cells and Induce Moderate ROS Increase, While Exerting Low Toxicity Towards Primary Cell Cultures: An In Vitro Study"

_ijms, 2025, doi:10.3390/ijms26114991_

Round 1

Reviewer 1 Report

Comments and Suggestions for Authors
  • please limit keywords to 5
  • please change "hah" in line 61
  • please pay attention to editorial errors and add spacebars e.g. line 65 t 40-50%[6], line 66 "disease. [7].", line 98, 103, 168 and more "described[14]
  • line 127, please change "(2.00 µM" to 2 µM
  • the micromoles marking in the drawings is unclear, please change the formatting
  • please present figures 1-3 as a data panel in one figure
  • all abbreviations in the figures should be explained in the figure legend
  • data from figure number 4 should be presented in the form of a data graph with statistical analysis
  • the same comment applies to figure 5
  • the reviewer does not understand why quite basic data on cell viability and IC50 values ​​are analyzed and presented in such an extensive way, dispersion graphs as in figures S1-S3. I think that the analysis from Figures 1-3 is sufficient. The whole analysis from the supplement does not contribute much and even distracts from the main part of the manuscript. The authors can mention in the manuscript that such an analysis was performed, but there is no point in describing it in separate chapters.
  • According to literature data, the DCFH probe detects not only hydrogen peroxide, but also the ROS production. Applications for DCFH include quantitation of oxygen-reactive species,  therefore the description of the results in section 2.3. should be changed
  • the results and discussion section (no. 2) lacks a real discussion of the results in the context of existing research, this section should be significantly improved, because in the vast majority of cases it is a description of the results and there is no discussion. Authors can also separate these two sections.
  • what is the purpose of describing already published data as a separate section in the submitted for review manuscript , namely "2.4. In Vitro Haemolytic Toxicity of BPPB on Red Blood Cells (RBC)"? This practice is not used. You can of course cite the publication but not add a chapter in the manuscript describing already published results.  Furthermore, scheme number 1 is identical to the one which is also scheme number 1 in the following paper: Alfei, S.; Torazza, C.; Bacchetti, F.; Signorello, M.G.; Passalacqua, M.; Domenicotti, C.; Marengo, B. Tri- 820 Phenyl-Phosphonium-Based Nano Vesicles: A New In Vitro Nanomolar-Active Weapon to Eradicate 821 PLX-Resistant Melanoma Cells. Int J Mol Sci 2025, 26, 3227, doi:10.3390/ijms26073227.
  • line 399 in vivo should be italicized
  • Spinal cord astrocytes and in cortical neurons were used as a control system to assess the biocompatibility of the tested compounds, therefore the results from Figure 7 should be combined in one panel with the results from Figure 1. Furthermore, correlation data e.g. from Figure 2 and 3 should also be combined in the form of a panel with the data from Figure 8 and 9
  • Why did the authors not evaluate cell death and ROS parameters for the control cell lines? 

Author Response

  • please limit keywords to 5

On request of this Reviewer, key words have been reduced to five.

  • please change "hah" in line 61

On request of this Reviewer, “hah” has been corrected in “have”. Please, see line 65 (revised version).

  • please pay attention to editorial errors and add spacebars e.g. line 65 t 40-50%[6], line 66 "disease. [7].", line 98, 103, 168 and more "described[14]

We thank the Reviewer for his/her comment. We already know this issue concerning the missing spacebars before citations, but it is not dependent on us. It depends on the use of Mendeley to insert citations, that sometimes does not allow the presence of spaces. If we correct, upon the reopening of the file, the issue occurs again. We have already experimented this undesired phenomenon, but in phase of creating the proof of manuscript, the Editorial Office will address this problem, as happened for other our manuscripts. We kindly ask the Reviewer to not considered it a duty for us.

  • line 127, please change "(2.00 µM" to 2 µM

We thank the Reviewer for his/her comment. Correction has been done at line 138 (revised version).

  • the micromoles marking in the drawings is unclear, please change the formatting

We thank the Reviewer for his/her comment. Following it, we have double-checked the word file on which we are working for revision without finding any problem with the micromolar marking. So, to clarify the question, we have opened the pdf version you should have received, and we found the problem you have signalled. The problem is not dependent on us, but it was born upon conversion of the word file in pdf file. Please, check the word file we have submitted to have confirmation. We will ask the Editorial Office to solve this problem.

  • please present figures 1-3 as a data panel in one figure

We thank the Reviewer for his/her suggestion. The operation asked has been done. Now only Figure 1 exists made of panels a, b, c, d, e and f. Figure 1 caption and the subsequent text up to Table 1 have been changed, accordingly. Figures numbering has been updated.

  • all abbreviations in the figures should be explained in the figure legend

Done. Please, check the caption of all Figures.

  • data from figure number 4 should be presented in the form of a data graph with statistical analysis
  • the same comment applies to figure 5

The images reported in old Figure 4 and 5, now Figure 2 and 3, were obtained by confocal microscope analyses. As the Reviewer certainly knows, this is a qualitative and not a quantitative analysis. Statistical analysis is possible only on results by quantitative tests. Precisely, to demonstrate the difference in labelling between treated and not treated cells, in both Figures, the same cells were observed and reported in bright field and in fluorescent field. Furthermore, to complete the information, merged images were also reported. On these results, observing Figure 4, in BPPB-treated HTLA, DAPI positive cells (labelled in blue) are in a greater number than those labelled in untreated cells and in treated HTLA ER cells and the same occurs in Figure 5.

  • the reviewer does not understand why quite basic data on cell viability and IC50 values ​​are analyzed and presented in such an extensive way, dispersion graphs as in figures S1-S3. I think that the analysis from Figures 1-3 is sufficient. The whole analysis from the supplement does not contribute much and even distracts from the main part of the manuscript. The authors can mention in the manuscript that such an analysis was performed, but there is no point in describing it in separate chapters.

Apologising in advance, we disagree with the Reviewer and explain why. The simple analysis of old Figures 1-3 (Figures 1a, 1b, 1c, 1d, 1e and 1f after revision) does not provide any evidence on the factors thru which BPPB exerts its cytotoxic effects: concentrations, exposure time, ROS overproduction? The same for the increase in ROS and ultimately for the cytotoxic effects of BPPB on primary cells. A concentration-dependent, time-dependent or ROS-dependent effect/mechanism has been often reported, without however verifying the accuracy of these assumptions analytically. Therefore, as already stated in the text in the unrevised version and now better clarified both in the abstract (lines 28-31 and 33,34) and at the end of the Introduction (lines 92-100), we have carried out an innovative analytical study to assess the possible existence of dependences between the cytotoxic effects of BPPB to cancer and normal cells and BPPB concentrations, the exposure time and the ROS increase. Such study was then extended also to the assessment of the possible existence of dependence between ROS increase and exposure timing or BPPB concentrations. To this end, proper dispersion graphs and their best fitting regression models were used, to analyse several series of data. With this approach, we have really established the existence or absence of a certain correlation between different couples of data, thus confirming or rebutting the presence of dependence between them. Since these operations are not so trivial, especially for researchers not expert in analytics, we thought it was necessary to both give due explanations in the main text and provide graphs and related equations, as well as the R2 values that justify our conclusions. We hope we have been exhaustive in giving the Reviewer the correct justification for our actions and we ask him to accept our choices. However, as previously mentioned, some parts have been added to the text to better highlight what has been done and the reasons/purposes for which it was done. Please, see also lines 270-285.

  • According to literature data, the DCFH probe detects not only hydrogen peroxide, but also the ROS production. Applications for DCFH include quantitation of oxygen-reactive species, therefore the description of the results in section 2.3. should be changed

We agree with the Reviewer and therefore the description of the results in section 2.3. has been modified as required. Please, see lines 360, 361, 370 and 371.

  • the results and discussion section (no. 2) lacks a real discussion of the results in the context of existing research, this section should be significantly improved, because in the vast majority of cases it is a description of the results and there is no discussion. Authors can also separate these two sections.

We thank a lot the Reviewer for this comment and valuable suggestion. The Reviewer is right concerning cytotoxic experiments and experiments to evaluate ROS induction by BPPB. Therefore, the discussion in Section 2 concerning these results has been extensively extended by reporting results by other Authors concerning cytotoxic data on cancer cells and ROS induction by similar and dissimilar nanoparticles. Please, see lines 171-217 and 382-402. Concerning discussion on cytotoxicity of BPPB to primary neurons and astrocytes, an extensive discussion was already present in the non-revised manuscript. You can find it in the revised manuscript in lines 542-552 and 564-589. Additionally, a discussion on the SI values, already existing in the original manuscript, can be found at lines 625-633 and 671-687.

  • what is the purpose of describing already published data as a separate section in the submitted for review manuscript , namely "2.4. In Vitro Haemolytic Toxicity of BPPB on Red Blood Cells (RBC)"? This practice is not used. You can of course cite the publication but not add a chapter in the manuscript describing already published results.  Furthermore, scheme number 1 is identical to the one which is also scheme number 1 in the following paper: Alfei, S.; Torazza, C.; Bacchetti, F.; Signorello, M.G.; Passalacqua, M.; Domenicotti, C.; Marengo, B. Tri- 820 Phenyl-Phosphonium-Based Nano Vesicles: A New In Vitro Nanomolar-Active Weapon to Eradicate 821 PLX-Resistant Melanoma Cells. Int J Mol Sci 2025, 26, 3227, doi:10.3390/ijms26073227.

We thank a lot the Reviewer for these indications. We have reduced Section 2.4 leaving only a sentence to indicate the reference (lines 473-474) and the same was done in the experimental Section 3.4 (line 758) Scheme 1 has been removed.

  • line 399 in vivo should be italicized

All in vivo and in vitro which were not italicized have been corrected (lines, 479, 500, 520, 772). We thank the Reviewer.

  • Spinal cord astrocytes and in cortical neurons were used as a control system to assess the biocompatibility of the tested compounds, therefore the results from Figure 7 should be combined in one panel with the results from Figure 1. Furthermore, correlation data e.g. from Figure 2 and 3 should also be combined in the form of a panel with the data from Figure 8 and 9

We apologise in advance to the Reviewer, but we make him/her kindly note that:

Figure 2 and 3 (not revised manuscript), Figure 1c, d, e and f do not contain correlation data as well as Figure 6a and 6b and 7a and 7b (revised manuscript). As reported in the text, Figure 1c and 1d (revised manuscript) contains the dispersion graphs of data reported as bars graphs in Figure 1a and b (neuroblastoma cells); Figure 6a and 6b (revised manuscript) contain the dispersion graphs of data reported as bars graphs in Figure 5a and 5b (revised manuscript) (primary cells); Figure 1e and 1f (revised manuscript) are the graphs of data conversion in Log 10 concentrations with related normalized responses of dispersion graphs reported in Figure 1c and d (neuroblastoma cells); Figure 7a and 7b (revised manuscript) are the graphs of data conversion in Log 10 concentrations with related normalized responses of dispersion graphs reported in Figure 6a and 6b (revised manuscript) (primary cells). Explained this, we think that the operation required by the Reviewer could create confusion and that Figures position should remain unchanged. That is, all results obtained for neuroblastoma cells should remain in the dedicated Section, as well as those obtained on primary astrocytes and neurons, to not generate confusion. Moreover, for more clarity on the different cytotoxicity of BPPB against tumour cell and reference normal cell, we had inserted in the manuscript, Figure 11 (now Figure 9), which graphically compare the results obtained on HTLA cells and primary ones, demonstrating the degree of biocompatibility in the form of selective index.

  • Why did the authors not evaluate cell death and ROS parameters for the control cell lines?

As reported in Table 2, the IC50 values of BPPB on primary cultures of cortical astrocytes and neuronal cells were approximately 7.044 and 1.56 µM, respectively. Apoptotic death in HTLA-230 cells and in HTLA ER cells was recorded in cells treated with 0.5 and 1 µM, respectively. Therefore, since the compound given at the concentrations capable of causing approximately 50% of cell death and an overproduction of ROS in HTLA-230 and in HTL ER cells were not cytotoxic for astrocytes and neurons, evaluating apoptosis and ROS in normal cells treated with these concentrations lower than the IC50 of the primary lines for comparison purposes, would not have led to any relevant results and would only have resulted in a waste of time and unnecessary additional expenses. These explanations have been included in the main text of the revised manuscript. You can find them in lines 589-598.

Reviewer 2 Report

Comments and Suggestions for Authors

The Remarkable and Selective In Vitro Cytotoxicity of Synthesized Bola-Amphiphilic Nanovesicles on Etoposide-Sensitive and -Resistant Neuroblastoma Cells

https://doi.org/10.3390/nano14181505

This study is similar to the present manuscript, the relevant data is the cytotoxic activity on the same Neuroblastoma cells lines on both studies. What is the different to previous study? that can contribute to the current study?

Please add in vivo study, to corroborate the efficiency. The apoptosis pathway should be determined in deeper. 

Author Response

https://doi.org/10.3390/nano14181505

This study is similar to the present manuscript, the relevant data is the cytotoxic activity on the same Neuroblastoma cells lines on both studies. What is the different to previous study? that can contribute to the current study?

Please add in vivo study, to corroborate the efficiency. The apoptosis pathway should be determined in deeper. 

We thank a lot the Reviewer for having spent his/her time in reviewing our manuscript and for his/her concise report.

Although this work could appear like https://doi.org/10.3390/nano14181505, it is not. Several are the points that make this new study dissimilar to the previous one and which make this study a notable advancement in our research about BPPB as novel anticancer option.

  • In this paper, for the first time the effects of BPPB on ROS production in HTLA 230 and HTLA ER cells has been carried out.
  • In this study, for the first time, an extensive innovative analytical study to assess the possible existence of dependence between the cytotoxic effects of BPPB to cancer and normal cells and BPPB concentrations, exposure time and ROS increase has been carried out. Such study was then extended also to the assessment of the possible existence of dependence between ROS increase and exposure timing or BPPB concentrations.
  • In this study, for the first time, proper dispersion graphs have been created, and their best fitting regression models were used to analyse several series of data. By doing so, we have really established the existence or absence of a certain correlation between different couples of data, thus confirming or rebutting the presence of dependence between them.
  • In this study, for the first time, we have demonstrated that BPPB is able to trigger apoptosis of both NB cell populations, and not necrosis, thus avoiding the emergency of secondary co-cancerogenic inflammation.
  • In this study, for the first time, the cytotoxic effects of the bola-amphiphilic nanovesicles of BPPB were evaluated on primary cell cultures of astrocytes and neurons.

Thinking of not forgetting other distinctive points that perhaps also exist, we hope that the Reviewer can be satisfied with these five ones.

Concerning the second point roused by the Reviewer, as in many other existing papers (only some examples https://doi.org/10.7717/peerj.8686, https://doi.org/10.4103/jpbs.jpbs_543_23, https://doi.org/10.7717/peerj.4358,  https://doi.org/10.1038/s41598-025-99841-9, https://doi.org/10.1016/j.sjbs.2021.05.004, Nawaz A, Riaz T, Ahmad A. In-vitro evaluation of antiproliferative potential of various fractions of Silybum marianum using HeLa and HepG2 cell lines. Pak J Pharm Sci. 2021 Mar; 34(2 (Supplementary)): 755-760. PMID: 34275811, https://doi.org/10.4103/0973-1296.191452, https://doi.org/10.4103/pm.pm_497_16), when a new synthesized compound is studied, the first step concerns the in vitro evaluations of its effects on target cells, in this case neuroblastoma cells (MDR and therapy-sensitive ones), and on healthy cells. Since several experiments and statistical analyses were performed to provide as much information as possible on the effects of BPPB and on the possible relationships between the effects (cytotoxicity and ROS production), concentrations and exposure time, the results obtained were considered exhaustive to propose this compound, as a promising possible therapeutic agent to treat MDR neuroblastoma. However, although we agree with the Reviewer that in vivo experiments need to be performed to corroborate in vitro findings, we felt that including in vivo studies in this already complex article would reduce its impact. Therefore, we believe that the next logical step of this research is the in vivo validation of the results obtained in vitro and this will be the aim of our next paper. Anyway, to better highlight that the aim of this study was to investigate the anticancer effects of BPPB in in vitro models, the title has been appropriately changed (lines 2-4), and other specifications have been added also in other parts along the manuscript. We kindly ask the Reviewer to understand our rational.

Round 2

Reviewer 1 Report

Comments and Suggestions for Authors

The authors have responded adequately to my questions and the paper can now be published.  

Author Response

The authors have responded adequately to my questions and the paper can now be published.  

We thank the Reviewer for his/her positive decision on our study.

Reviewer 2 Report

Comments and Suggestions for Authors

In the current form the manuscript cannot be accepted for publication, as it resembles closely a previously published research paper by the authors titled “The Remarkable and Selective In Vitro Cytotoxicity of Synthesized Bola-Amphiphilic Nanovesicles on Etoposide-Sensitive and -Resistant Neuroblastoma Cells” https://pmc.ncbi.nlm.nih.gov/articles/PMC11434613/#sec2-nanomaterials-14-01505.

Such published paper had the novelty of reporting for the first time the Bola-Amphiphilic Nanovesicles (BPPB) synthesis and their effect on cell viability, however, the present manuscript is too similar in justification and results, therefore, it lacks impact and relevance.

It is also important to highlight that the manuscript and the responses given by the authors states that the novelty of this paper is that BPPB induces apoptotic death, however, the methods used (annexin V assay, and simple DAPI staining), as well as the evidence provided are insufficient to support such statement. The ROS assays are also not well justified; thus, they do not provide relevant information to the study.

Major methodological flaws are also present; the cell lines are not well described; the authors do not consider other methods to assess cell viability and whether the methods used are the most appropriate for the study and the research question to be answered.

The authors also state that the primary cell cultures are cortical neurons and spinal cord astrocytes, however, there is no description about how the cells were confirmed to be such cell type (markers, cell sorting, purification).

For the manuscript to be accepted the authors should make notable changes and deepen into the actual apoptosis pathways. They should use appropriate and well described assays and markers to confirm that the BPPB induces cell death by such mechanism.

Author Response

In the current form the manuscript cannot be accepted for publication, as it resembles closely a previously published research paper by the authors titled “The Remarkable and Selective In Vitro Cytotoxicity of Synthesized Bola-Amphiphilic Nanovesicles on Etoposide-Sensitive and -Resistant Neuroblastoma Cells” https://pmc.ncbi.nlm.nih.gov/articles/PMC11434613/#sec2-nanomaterials-14-01505.

Such published paper had the novelty of reporting for the first time the Bola-Amphiphilic Nanovesicles (BPPB) synthesis and their effect on cell viability, however, the present manuscript is too similar in justification and results, therefore, it lacks impact and relevance.

It is also important to highlight that the manuscript and the responses given by the authors states that the novelty of this paper is that BPPB induces apoptotic death, however, the methods used (annexin V assay, and simple DAPI staining), as well as the evidence provided are insufficient to support such statement. The ROS assays are also not well justified; thus, they do not provide relevant information to the study.

Unequivocally, this Reviewer is very concerned about the novelty, originality and relevance of this new study, since she/he is fully convinced that no new relevant information has been reported respect to our previous article, she/he cited. Based on the Reviewer's comments and requests, we regret to ascertain that she/he probably does not appreciate our study and does not recognise its novelty, and this is her/his right, but we believe that he/she can appreciate our effort to further explain the novelty of this article. In fact, in addition to ROS monitoring and annexin V assay associated to DAPI staining, other relevant experiments and novel approaches to better interpret the results have been carried in this study for the first time. In detail, by an innovative analytical approach, we have investigated and determined if the cytotoxic effects of BPPB towards cancer and normal cells could depend on BPPB concentrations, exposure times and/or ROS overproduction. Such study was then extended also to assess if ROS increase could depend on exposure timing and/or BPPB concentrations. In this study, for the first time, proper dispersion graphs have been created, and their best fitting regression models were used to analyse several series of data. By this approach, we have established the existence or absence of a certain correlation between different couples of data, thus confirming or rebutting the presence of dependence between them. Specifications on these questions have been added along the abstract (lines 27-33) and at the end of Introduction (93-107). Here, we can already provide the Reviewer first evidence that ROS overproduction monitoring by the DCFH essay, commonly used to this scope, provided relevant information to the study, despite not detected by the Reviewer. In fact, it allowed to demonstrate the existence of a certain ROS-dependent mechanism governing the cytotoxic effects of BPPB in turn depending on time of exposure and type of cell line (lines 31-33, Section 2.3.3). Additionally, we have decided to measure ROS levels since HTLA ER cells developed a strong resistance to etoposide and other chemotherapeutics, that exert their cytotoxic action by stimulating ROS overproduction, as a consequence of the increase in their antioxidant defences (https://doi.org/10.18632/oncotarget.12209). In this study, the evaluation of ROS levels in cells treated with BPPB was particularly useful to understand whether i) this nanovesicle-based preparation exerted its cytotoxic action through ROS production (this was confirmed); ii) the resistance acquired following chronic treatment with etoposide also conferred resistance to a preparation based on the use of nanovesicles (this was confuted). Moreover, the evaluation of ROS levels can be an indirect method to ascertain the presence of apoptosis (low levels of ROS induce apoptosis and high levels necrosis) (https://doi.org/10.1089/ars.2007.1957; doi.org/10.1023/A:1009616228304). Specification on the importance and relevance of ROS monitoring have been included also in the main text. Please, see lines 360-369. The suitability of test used to monitor ROS production was instead included in lines 372-376.

Going on about the novelty and relevance of this study, the cytotoxic effects of the bola-amphiphilic nanovesicles of BPPB were here evaluated for the first time on primary spinal cord astrocytes and cortical neurons cultured in our laboratories starting from mice. The Reviewer should consider that only these last experiments could alone make our study robust, relevant and new compared to the previous one. In fact, while it is common that new compounds with antitumor activity are tested on tumour cell lines, such as Hela, HepG2, HaCat, COS-7 etc. (as in our previous works), to investigate their biocompatibility, it is difficult that primary cell cultures are used. Furthermore, as highlighted in the text, no one before us, had tested bola-amphiphiles like our BPPB on primary cultures of neurons and astrocytes. Finally, regarding the Reviewer’s concern about our choice to assess apoptosis by labelling cells with Annexin-V-FITC and propidium iodide, we regret that the Reviewer does not agree with us, since our choice is justified by numerous papers published in high-impact journals. In fact, this method is a standard procedure to monitor the progression of apoptosis. In fact, early apoptotic cells are Annexin V-positive and PI-negative (Annexin VFITC1/PI2), whereas late (end-stage) apoptotic cells are Annexin V/PI-double-positive (Annexin V-FITC1/PI1).

The assay is based on the evidence that the cell plasma membrane is asymmetric, and that phosphatidylserine (PS) is normally restricted to the inner leaflet of the plasma membrane. As surely the Reviewer know, apoptosis is characterized by the loss of lipid asymmetry and PS becomes exposed on the outer leaflet of the plasma membrane. Then, Annexin V binds to PS exposed on the outside and fluorescently labelled Annexin V can be used to detect apoptotic cells. Annexin V can also stain necrotic cells because these cells have ruptured membranes that permit Annexin V to access the entire plasma membrane. However, apoptotic cells can be distinguished from necrotic cells by co-staining with propidium iodide (PI) because PI enters necrotic cells but is excluded from apoptotic cells. [Crowley LC, ett al., Cold Spring Harb Protoc. 2016 Nov 1;2016(11). doi: 10.1101/pdb.prot087288; Wlodkowic D, et al., Methods Cell Biol. 2011;103:55-98. doi: 10.1016/B978-0-12-385493-3.00004-8; Sawai H, et al., Biochem Biophys Res Commun. 2011 Aug 5;411(3):569-73. doi: 10.1016/j.bbrc.2011.06.186; Liu T, et al., Anal Chem. 2009 Mar 15;81(6):2410-3. doi: 10.1021/ac801267s. Kumar R, et al., Methods Mol Biol. 2021;2279:213-223. doi: 10.1007/978-1-0716-1278-1_17. PMID: 33683697]. Specifications on this question have been included also in the main text. Please, see lines 226-227.

Major methodological flaws are also present; the cell lines are not well described;

We thank the Reviewer for his/her suggestion. As required HTLA 230 and HTLA ER were better described in the Materials and Methods Section (Section 3.2.1) and additional references have been added. Please see lines 739-753.

the authors do not consider other methods to assess cell viability and whether the methods used are the most appropriate for the study and the research question to be answered.

We thank the Reviewer for this comment, which enabled us to give clearer explanations concerning the reliability and suitability of the methods used in this study and to justify our choices. Concerning NB cells, cell viability was determined by using the CellTiter 96® AQueous One Solution Cell Proliferation MTS Assay. It is a colorimetric method commonly used for determining the number of viable cells in proliferation, their chemosensitivity and the cytotoxic effects of a new chemical under early investigation on different cell lines, as it was in our scope. We selected this method, because it represents an eco- and user-friendly tool suitable for our scope. As reported in several papers, MTS assay is time efficient and safe (https://doi.org/10.1016/S0300-483X(97)00151-0). MTS is commonly used by researchers investigating cancer cells, to understand how specific treatments influence growth inhibition and viability. The MTS assay allows to evaluate cellular health and response to therapeutic agents. Collectively, the MTS cell proliferation assay stands as a cornerstone in modern biological research and can enable advancements in cancer studies, as in our case, but not only, affirming its status as an indispensable tool (https://doi.org/10.1016/S0300-483X(97)00151-0, https://labverra.com/articles/mts-cell-proliferation-assay-insights-applications/). Please, see lines 117-121 and 766-769.

Concerning primary cell cultures, the cytotoxic properties of BPPB were assessed using the MTT (3-(4,5-dimethylthiazol-2-yl)-2,5-diphenyltetrazolium bromide) assay, following the manufacturer’s protocol. The MTT assay, similarly to the previously described MTS, is a colorimetric test based on the measurement of cell metabolic activity, which reflects the number of viable cells present. Although we agree with the Reviewer observation about the possibility of using alternative and complimentary cytotoxicity tests, such as the LDH assay or the ATP assay, the  MTT assays is still a method widely accepted for this scope, with decades of use in thousands of publications (https://dx.doi.org/10.1101/pdb.prot095505). Indeed, is commonly used to measure cytotoxicity (loss of viable cells) or cytostatic activity (shift from proliferation to quiescence) of potential medicinal agents, thus resulting particularly suitable for our study (https://dx.doi.org/10.1101/pdb.prot095505). Moreover, MTT essay was chosen for its simplicity, convenience, quantitative and high throughput. It is a method widely accepted for this scope, with decades of use in thousands of publications (https://dx.doi.org/10.1101/pdb.prot095505). MTT assay is a well-established method for assess cell viability and its reliability is supported by standardized protocols in many areas of research. Additionally, it is cost-effective, and its results are objective and reproducible (https://acmeresearchlabs.in/2025/02/27/mtt-cytotoxicity-assay-lab-principle-protocol-applications/). Please, see lines 536-541.

Specification on this question have been included in the revised text at the indicated lines.

The authors also state that the primary cell cultures are cortical neurons and spinal cord astrocytes, however, there is no description about how the cells were confirmed to be such cell type (markers, cell sorting, purification).

We thank the Reviewer for giving us the opportunity to explain our point of view. Our research group is mainly focused on neuroscience studies using in-vivo and in-vitro models. We were honoured to give our contribution to this paper using primary cell cultures. We believed that testing BPPB on CNS primary cell cultures could give an adding value to the article. Concerning spinal cord astrocytes primary cell cultures, we are conscious that they are not a common model used for in-vitro studies, because of its difficult method of isolation and culture, although we believe it represents a very useful and complementary model in parallel to neurons. We extensively characterised the nature and purity of these cultures in previous papers that have now been cited in the method section (lines 811-812). We believe it is out of the scope of the paper to introduce new results to confirm the characterisation of this cell type.

Concerning cortical neurons primary cell cultures instead, they represent a very common model widely used for different in vitro studies, including cytotoxicity (Schmuck et al., 2000, PMID: 10900406; Jurič DM, et al., 2022 PMID: 36287988). For this reason, we considered totally not relevant to further demonstrate and confirm, in the present paper, the nature and purity of this cell type. We now introduce one of our recent publications using this model, however, cortical neurons are for similar studies. Please, see line 824.

For the manuscript to be accepted the authors should make notable changes and deepen into the actual apoptosis pathways. They should use appropriate and well described assays and markers to confirm that the BPPB induces cell death by such mechanism.

Now this Reviewer should consider that, as understandable by our responses to previous points, the suitability, reliability and robustness of methods used by us to investigate NB cells death mechanisms and monitoring ROS overproduction induced by BPPB, have been demonstrated, also providing several literature evidence. We are aware the manuscript can be always improved by adding further studies, assays, markers or proof-of-concept experiments, however we believe that the Reviewer will understand that we did our best and effort to ameliorate suitability, reliability and robustness of methods used to investigate BPPB effects. Concerning apoptosis, we agree with the Reviewer that the apoptotic pathway has not still in deeply investigated here, but it was not in the scope of this study. We, for the moment, wanted only to give raw but precious information regarding which kind of death, apoptotic or necrotic, BPPB induces, and apoptosis development, since if substantial necrosis would be observed, further investigations on BPPB, would have been drastically compromised. We prompt the Reviewer to consider that literature is full of works on the antitumor activity of new active chemicals, where there is not even the slightest indication of apoptosis and necrosis, which, despite being considered relevant. We could understand the Reviewer's determination in asking for more in deep investigation on apoptosis pathway to detect markers etc. that confirm our assumption, if the journal we chose to publish our study was a specialized journal on cancer, but this is not the case. Concerning ROS overproduction, the relevant information provided by their monitoring upon treating NB cells with increasing concentrations of BPPB for increasing time exposure, as made in experiments to assess cell viability, was already evidenced in the not-revised manuscript, starting from the abstract. Anyway, upon this second round revision, further evidence has been provided on the relevance and utility of ROS essay, both to the Reviewer and in the main text, in specific parts which have been indicated to the Reviewer. Thank to our novel analytical approach used to investigate possible correlations between the events investigated, and ROS essay, we have for example unveiled the existence of a ROS-dependent mechanism in 24-hours and 24/48-hours treatments of HTLA ER and HTLA 230, respectively. Moreover, as the Reviewer can appreciate, notable changes have been made in the manuscript according to her/his requests. Specifically, NB cells have been extensively described, and general justifications for our choices concerning the methods used to assess cell viability have been provided to the Reviewer and included in the text. Finally, confirmations and literature evidence that neurons and astrocytes cultured and isolated by us and used in this study are exactly those intended cells have been provided to the Reviewer and in the text. On these considerations, we kindly ask the Reviewer to understand our effort and better consider our study endorsing the publication.

Round 3

Reviewer 2 Report

Comments and Suggestions for Authors

After reviewing the manuscript and response to the reviewer; there are still major concerns that make the work not appropriate for publication yet.

Science should be based on the proper and objective analysis of the results rather than simple backing up on literature review. Overall, the scientific method is not well applied to the study, since the affirmations the authors make are not sustained by the results obtained.

The authors raised a concern when asked to deepen their work in the mechanisms involved in apoptosis and back up the assumption they state in the title “TPP-Based Nanovesicles able to Induce Apoptotic Death in Multidrug Resistant Neuroblastoma Cells exert a Low toxicity Towards Primary Cell Cultures”.

They reply that “We could understand the Reviewer's determination in asking for more in deep investigation on apoptosis pathway to detect markers etc. that confirm our assumption, if the journal we chose to publish our study was a specialized journal on cancer, but this is not the case”.

However, the authors have summited their manuscript to the International Journal of Molecular Sciences, in the Section Molecular Oncology and the Special Issue New Molecular Mechanisms and Advanced Therapies for Solid Tumors impact factor 4.9.  For the work to make a contribution to the field and provide new knowledge it is important that the authors meet the standards and confirm the molecular mechanism involved, not only a non quantified general apoptosis assay and total ROS.

Although the importance of this works should reside in the idea that the new BPPB compound induces cell death, this should be well sustained.

Among the various revisions found:

-The fact that a linear regression is not a correlation and should not be presented as such.

-As the literature cited by the authors for a correct Annexin/PI analysis this assay is performed by flow cytometry, for proper and quantifiable result. Not just an image.

-DAPI staining is not indicative of necrosis and is unacceptable to be written off as such.

-The authors state that their mechanism is ROS-dependent, however, this is highly inaccurate, as similarly to the other experiments it is not well design. The experiment lacks a positive control, the highest value is around 6%, and statistical difference does not equal biological effect.

-The reuse of data in “In Vitro Haemolytic Toxicity of BPPB on Red Blood Cells (RBC) 487 Haemolytic toxicity of BPPB (HC50 = 15.56 ± 12.13 µM) was assessed in our recent 488 study [13].” Not only is inappropriate but it also does not add value.

-Please confirm the identity of the primary cell culture (Markers).

It is also necessary to highlight that upon close and detailed inspection there is an alarming amount self and college citations

11,12,13,14,15,16,24,46,61,62,64,65,66,67,69 that can be identified.

In general, strongly recommended a good experimental strategy to resolve the mechanism involved with cellular death induced by its molecule

Author Response

After reviewing the manuscript and response to the reviewer; there are still major concerns that make the work not appropriate for publication yet.

We can only answer this Reviewer, that we had no doubts. Once, another Reviewer, whose professionalism and high capacity of judgment I recognize, pointed out to me that the process of reviewing a manuscript is not a fight between two combatants, who try to win at the expense of the other, but it must be a productive exchange of opinions and criticisms between scientists, aimed at improving the work, in the interest of both and of the scientific community. This is certainly not the case, since despite our attempts to satisfy Reviewer requests, her/his report on the system about Introduction, research design appropriability, methods description, results presentation and conclusions is paradoxically getting worse, respect to the previous times. Either this Reviewer is getting more and more annoyed because we have always responded to all her/his comments or she/he wants to deliberately sink our work.

Science should be based on the proper and objective analysis of the results rather than simple backing up on literature review. Overall, the scientific method is not well applied to the study, since the affirmations the authors make are not sustained by the results obtained.

We disagree with the Reviewers on this point. As already stated in the previous round revisions, the methods used for our experiments are universally recognized and have been used in several articles, ours included, published on high-impact journals. They have produced quantitative and qualitative results, depending on the type of investigation carried out. We well know that when we have provided only qualitative information, quantitative one could be possible, but not everything can be done in one job. The assertions and affirmations reported in our paper, such as IC50 towards both cancer and normal cells, ROS%, selectivity index values, presence of early and late apoptosis by confocal qualitative analysis, correlations between the events investigated etc. have been directly deduced by the observed results, which unequivocally support them.

The authors raised a concern when asked to deepen their work in the mechanisms involved in apoptosis and back up the assumption they state in the title “TPP-Based Nanovesicles able to Induce Apoptotic Death in Multidrug Resistant Neuroblastoma Cells exert a Low toxicity Towards Primary Cell Cultures”.

If Title in this form seems too risky to the Reviewer we can change it. So, to assure the scientific conscience of Reviewer, the title has been changed to “TPP-Based Nanovesicles Kill MDR Neuroblastoma Cells and Induce Moderate ROS Increase, While Exert Low Toxicity Towards Primary Cell Cultures: an in Vitro Study”. By this way, we have deleted the information that seems misleading and premature to Reviewer, without further experimental confirmations from quantitative analysis of the apoptosis markers, she/he requests. Please, see lines 2-4. The further step of our research, required by Reviewer was not in the scope of this study. We kindly ask for the third time to Reviewer to try to accept our design.

They reply that “We could understand the Reviewer's determination in asking for more in deep investigation on apoptosis pathway to detect markers etc. that confirm our assumption, if the journal we chose to publish our study was a specialized journal on cancer, but this is not the case”.

However, the authors have summited their manuscript to the International Journal of Molecular Sciences, in the Section Molecular Oncology and the Special Issue New Molecular Mechanisms and Advanced Therapies for Solid Tumors impact factor 4.9.  For the work to make a contribution to the field and provide new knowledge it is important that the authors meet the standards and confirm the molecular mechanism involved, not only a non-quantified general apoptosis assay and total ROS.

We well know that Reviewer does not like that we provide literature examples. Anyway, to smooth the Reviewer’s concern about meeting the standards of IJMS, we can provide her/him evidence of other our articles on cancer published by us in IJMS not containing neither the qualitative experiments on apoptosis carried out by us in this study nor the quantitative experiments asked by the Reviewer. Nevertheless, they have found space in IJMS.

https://doi.org/10.3390/ijms26073227, https://doi.org/10.3390/ijms251810071, https://doi.org/10.3390/ijms241915027

In addition, we can provide evidence of other articles on cancer, already published by other authors in IJMS, not containing neither qualitative or quantitative experiments, to detect apoptosis and apoptosis markers, or containing only qualitative experiments like those carried out by us in this paper. These manuscripts were however considered in compliance with IJMS standards and accepted for publication.

https://doi.org/10.3390/ijms9081407, https://doi.org/10.3390/ijms9050864, https://doi.org/10.3390/ijms9050821, https://doi.org/10.3390/ijms141223315, https://doi.org/10.3390/ijms140815755  

Since this time, this Reviewer has also commented on the Special Issue (SI) we have selected for our work, saying that such SI calls for papers in which molecular mechanisms should be studied more in deep than we have made in our work, we asked the Editors of IJMS to decide to transfer our paper to another more appropriate SI. This Reviewer could her/himself suggest a possible SI, if she/he wants.

Although the importance of this works should reside in the idea that the new BPPB compound induces cell death, this should be well sustained.

This comment of the Reviewer left us astonished. There is all a Section (Section 2.2), which demonstrates that BPPB induces cell death in NB cells. There are both bar and dispersion graphs, evidencing how cell viability dramatically decreased upon administration of increasing concentrations of BPPB for three different times of exposure, as well as the very low IC50 values, calculated, using adequate software and method, for BPPB on both cancer cells populations considered in this study. They are clearly reported in Table 1, page 5. What else do we need to do, according to the Reviewer, to demonstrate that BPPB kills tumour cells? We apologize in advance to Reviewer for our ignorance, but these are usually the demonstrations which are provided to prove that there was cell death. Additionally, as we said in the previous round revision, the MTS assay used by us to obtain this information is universally recognized to investigate cell viability and therefore to evaluate the ability of a compound to kill cells.

Among the various revisions found:

-The fact that a linear regression is not a correlation and should not be presented as such.

Where has the Reviewer found this information?

We are confident to assert that when a linear regression model can be constructed using couples of data variables (x, y) of a dispersion graph, it just expresses a correlation (in this case linear) between the two series of data. Obviously, not always when a straight line is passed through a set of points usually using the least squares method as excel software makes, this linear correlation exists. It can be established, based on the value of R (not by chance called correlation coefficient) or better on that of R2 (determination coefficient) which should be > 0.95. Anyway, also other types of correlations depending on the existence of nonlinear regression fitting couples of variables could exist, always on the base of R2 values, as observed in our study. In this regard, we suggest the Reviewer to consider what reported at link https://openstax.org/books/principles-data-science/pages/4-3-correlation-and-linear-regression-analysis, to have confirmation of our explanation.

-As the literature cited by the authors for a correct Annexin/PI analysis this assay is performed by flow cytometry, for proper and quantifiable result. Not just an image. DAPI staining is not indicative of necrosis and is unacceptable to be written off as such.

We recognize that literature articles previously provided by us used Annexin/PI analysis in association with flow cytometry, but we have provided those articles, since, last time, the Reviewer concerns regarded the suitability of the type of essay (Annexin/PI) and not of the method (fluorescence or flow cytometry) used to observe and present results. If now the problem is this, we can assert that also just images by confocal fluorescence microscopy are accepted for a correct Annexin/PI analysis. We have infact found reported that, “Morphological features observed under fluorescence or confocal microscope after Hoechst 33,342, acridine orange (AO), or 4',6-diamidino-2-phenylindole (DAPI) staining, which indirectly reveals the nuclear and chromatin conditions of the cells, according to the intensity and distribution of the fluorescence signals, determine the occurrence of apoptosis” (https://doi.org/10.1080/15384101.2021.1919830). Another example is in https://doi.org/10.1007/s11356-020-08191-8. These new references have been included in the main text, in line 220.

-The authors state that their mechanism is ROS-dependent, however, this is highly inaccurate, as similarly to the other experiments it is not well design. The experiment lacks a positive control, the highest value is around 6%, and statistical difference does not equal biological effect.

Perhaps this Reviewer has not clearly understood what value has been reported on y axis. It is not ROS% but the % of cells positive to DCFH normalized for cell proteins content. So, 6% in our graphs does not mean 6% ROS, but a ROS overproduction 6 times higher than in control cells, where ROS production was fixed to 1%. We specify that raw fluorescence data is in fact not applicable since an increase of fluorescence could be due to a high number of cells but also to an increased production of H2O2 induced by the treatments. This is the reason why it is necessary to normalize the fluorescence data to the protein content. This Reviewer can may object that these details were not clearly specified in this work, but they were detailed in other works of us, we cited. However these specifications have now been inserted both in the Discussion (lines 365-369), in the Experimental part (lines 781-787) and in the caption of Figure 4 (lines 378-380).

-The reuse of data in “In Vitro Haemolytic Toxicity of BPPB on Red Blood Cells (RBC) 487 Haemolytic toxicity of BPPB (HC50 = 15.56 ± 12.13 µM) was assessed in our recent 488 study [13].” Not only is inappropriate but it also does not add value.

We agree with this Reviewer. The sections specifically dedicated to the haemolytic toxicity in this paper have been removed. The section numbering has been updated.

-Please confirm the identity of the primary cell culture (Markers).

We have addressed Reviewer request in lines 812-825 and 852-861.

It is also necessary to highlight that upon close and detailed inspection there is an alarming amount self and college citations

11,12,13,14,15,16,24,46,61,62,64,65,66,67,69 that can be identified.

According to IJMS requirements, self- and colleague-citations should be under 20%. Since in our paper they account for the 19.7%, our study is in compliance with the requirements of IJMS.

In general, strongly recommended a good experimental strategy to resolve the mechanism involved with cellular death induced by its molecule

It was not in the scope of this study.